# Using deep learning to quantify neuronal activation from single-cell and spatial transcriptomic data

Ethan Bahl[1,2], Snehajyoti Chatterjee [3,4], Utsav Mukherjee [3,4,5], Muhammad Elsadany [1,2], Yann Vanrobaeys [2,3], Li-Chun Lin[3,4], Miriam McDonough [4,6], Jon Resch[4], K. Peter Giese [7], Ted Abel [3,4] & Jacob J. Michaelson [1,3,8,9]

Neuronal activity-dependent transcription directs molecular processes that regulate synaptic plasticity, brain circuit development, behavioral adaptation, and long-term memory. Single cell RNA-sequencing technologies (scRNAseq) are rapidly developing and allow for the interrogation of activity-dependent transcription at cellular resolution. Here, we present NEUROeSTIMator, a deep learning model that integrates transcriptomic signals to estimate neuronal activation in a way that we demonstrate is associated with Patch-seq electrophysiological features and that is robust against differences in species, cell type, and brain region. We demonstrate this method's ability to accurately detect neuronal activity in previously published studies of single cell activity-induced gene expression. Further, we applied our model in a spatial transcriptomic study to identify unique patterns of learning-induced activity across different brain regions in male mice. Altogether, our findings establish NEUROeSTIMator as a powerful and broadly applicable tool for measuring neuronal activation, whether as a critical covariate or a primary readout of interest.

Transcriptional regulation is a key component in the downstream response to neuronal stimulation, and the dysregulation of this activity-dependent gene expression has been linked to several nervous system disorders[1–6]. Although several techniques exist to identify active neurons with transgenic single-gene reporter systems[7–10], there is currently no method available to estimate neuronal activity from transcriptome-wide signal in gene expression data. Estimating neuronal activation based on gene expression data is vital for unraveling the intricate processes governing neuronal function and dysfunction, and decades of research has been devoted to characterizing these processes[1,4,11]. Accurate estimation of activation could provide insights into neural circuitry, advance our understanding of neurological disorders, and ultimately contribute to the development of targeted therapeutic interventions. With advances in single-cell RNA-sequencing (scRNAseq) technologies, researchers can now study activity-dependent transcription at the individual neuron level[12,13]. However, scRNAseq has persistent limitations such as sparsity and stochasticity in gene expression measurements, which challenge our ability to use these tools to understand neuronal activation[14–17].

[1]Department of Psychiatry, University of Iowa, Iowa City, IA, USA. [2]Interdisciplinary Graduate Program in Genetics, University of Iowa, Iowa City, IA, USA. [3]Iowa Neuroscience Institute, University of Iowa, Iowa City, IA, USA. [4]Department of Neuroscience & Pharmacology, University of Iowa, Iowa City, IA, USA. [5]Interdisciplinary Graduate Program in Neuroscience, University of Iowa, Iowa City, IA, USA. [6]Interdisciplinary Graduate Program in Molecular Medicine, University of Iowa, Iowa City, IA, USA. [7]Department of Basic and Clinical Neuroscience, King's College London, London, UK. [8]Department of Biomedical Engineering, University of Iowa, Iowa City, IA, USA. [9]Department of Communication Sciences & Disorders, University of Iowa, Iowa City, IA, USA. ✉ e-mail: jacob-michaelson@uiowa.edu

To address these challenges, we present an innovative tool that estimates neuronal activation using transcriptome-scale data as input. Using a comprehensive multi-species gene expression dataset, our deep learning model employs a single unit bottleneck to derive interpretable neuronal activity scores and an auxiliary input to the decoder to counteract dataset-specific biases. This novel approach, which distills the whole transcriptome into a single integrative activity score, challenges the single-gene paradigm (e.g., *Fos*) for measuring activity at the single cell and bulk tissue level.

Distilling transcriptome-scale signatures into a one-dimensional information bottleneck, our approach reconstructs the expression profiles of 22 well-established neuronal activity markers. We show that the resulting activity score is significantly associated with electrophysiological features of increased excitability, demonstrating its applicability in classifying neurons along this essential characteristic. We further show its application in secondary analyses of several single-cell data sets, as well as in new spatial transcriptomic data that captures learning-related circuit activation in the murine brain.

To facilitate community usage, we developed NEUROeSTIMator, an R package with a tutorial that demonstrates our model's application to single-cell data. This innovative method offers a powerful solution to estimate neuronal activation based on de-noised latent patterns in single-cell gene expression, outperforming alternative approaches and providing significant associations with electrophysiological features at the single-cell level.

## Results

### Multi-species training, test, and validation data sets

We developed a neural network to quantify signatures of transcriptional activation by training an autoencoder to reconstruct expression of 22 coexpressed activity-dependent genes from a 1-dimensional latent space (Fig. 1). We trained, validated, and tested the model using publicly available datasets, including single cell datasets from the Allen Institute of Brain Science[18] (AIBS) totaling over one million combined samples, as well as GEO datasets and a novel spatial transcriptomics dataset (Supplementary Data 1). Activity-dependent genes were identified by intersecting differential expression results of three studies of experimental manipulation of neuron activity[19–21]. All three differential expression studies examined different brain regions, used different methods of neuronal stimulation, and were published from independent groups. Intersecting these three lists yielded a list of 29 candidate target genes (i.e., genes to be reconstructed in autoencoder output) Principal component analysis of the candidate genes revealed a relatively low dimensionality of expression profiles, with PC1 explaining a majority of the variance (Supplementary Fig. 1a). Using the Allen datasets, we then performed bootstrap permutations of principal component analysis on candidate target genes and found a subset of 22 genes robustly exhibiting greater PC1 loadings than expected under a null PCA with randomly shuffled data (Supplementary Fig. 1b). These 22 genes were selected as the final set of target genes to be reconstructed by the autoencoder during training. The Allen Cell Type Databases were downsampled with hierarchical weighting to increase

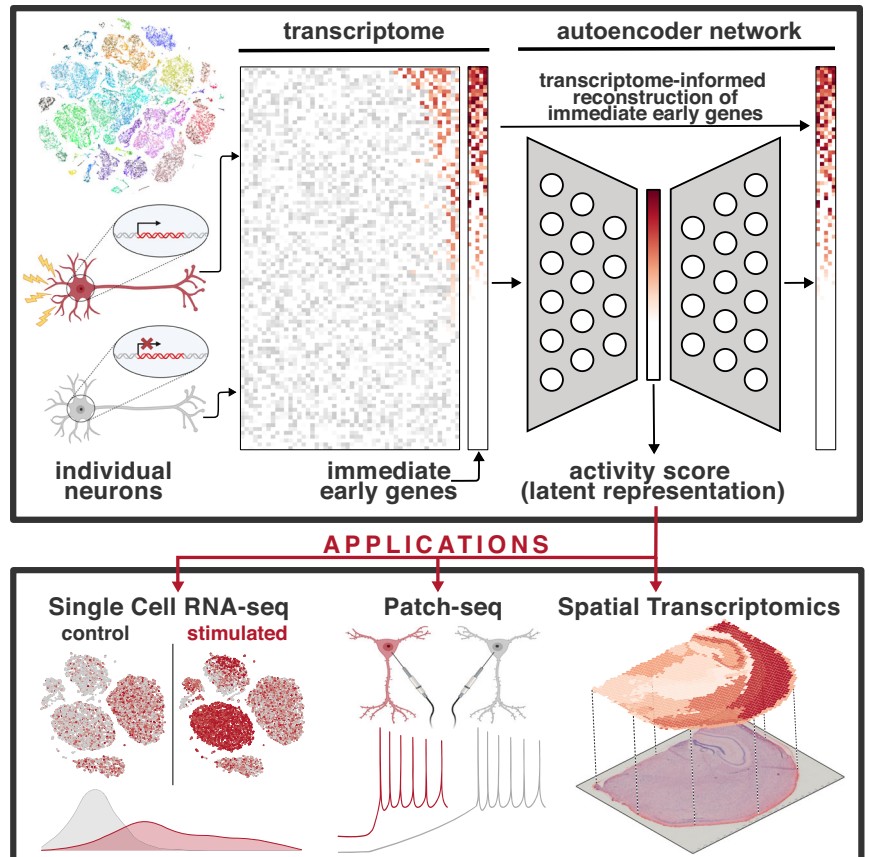

**Fig. 1 | Schematic Overview of NEUROeSTIMator.** The graphical abstract illustrates the application of NEUROeSTIMator, a deep learning model, to estimate neuronal activity. Transcriptome-wide gene expression data from individual neurons is input to an autoencoder, which reconstructs expression of immediate early genes from a single unit latent space called the 'activity score'. The activity score can be applied to gene expression data, including single cell RNA-sequencing, Patch-seq, or spatial transcriptomics studies for detecting transcriptomic signatures of activity.

balance of cell types, datasets, technical variation, and biological variation (Supplementary Fig. 2a, b). Five hundred thousand neurons were selected and partitioned into cell type-balanced training and testing sets, and the training set was further split into 5 folds for cross validation. We trained the neural network to predict expression of the 22 activity-dependent target genes through a 1-dimensional hidden bottleneck layer with sigmoidal activation (see methods for a detailed description of model architecture and training). To evaluate model performance, we applied it to a diverse test set of 164,658 neurons held out from the training process, including 86,650 samples from the Allen and 78,108 samples from additional labeled datasets. We selected our final model based on classification accuracy of labeled test set data through cross validation.

For further analyses, model output at the bottleneck activation layer was extracted to index activity level for each neuron passed through the model. Hereafter, we refer to this output as the predicted activity or activity score.

## Genes informing model predictions

To identify genes whose expression levels influence model predictions, we calculated integrated gradients[22] for all input genes with respect to predicted activity using data from the held out test set. Integrated gradients attribute model predictions to input features for each cell. We first compared the attributions of target and non-target genes on predicted activity. The most influential genes were *Fos, Egr1, Fosl2, Junb, Fbxo33*, and *Nr4a1*. We further explored gene importance for all non-target input genes and found varying degrees of influence throughout the transcriptome (Supplementary Data 2). The most influential non-target genes, such as *Fosb, Npas4, Egr2*, and *Ntrk2* tended to have higher initial support for consideration as a target gene and were more often differentially expressed in multiple transcriptomic studies of neuronal activity-dependent transcription (Supplementary Fig. 3a). Other known activity response genes such as *Homer1, Egr4, Dusp6*, and *Bdnf* were also among the top fifty genes with the highest feature attribution. To evaluate whether influential genes were enriched for specific annotated biological mechanisms, we performed gene set enrichment analysis on gene ontology and pathway annotation sets (Supplementary Fig. 3b, Supplementary Data 3). Among the most highly enriched gene sets were DNA binding transcription factor activator activity ($p_{adjusted} = 3.97 \times 10^{-9}$, NES = 2.29), signaling by NTRKs ($p_{adjusted} = 2.07 \times 10^{-8}$, NES = 2.40), cognition ($p_{adjusted} = 5.35 \times 10^{-3}$, NES = 2.01), MAPK signaling pathway ($p_{adjusted} = 2.02 \times 10^{-4}$, NES = 2.01), circadian rhythm ($p_{adjusted} = 1.60 \times 10^{-3}$, NES = 2.03), and cellular response to extracellular stimulus ($p_{adjusted} = 0.033$, NES = 1.89). To identify gene sets with higher attribution in non-target genes, we removed target genes, repeated the analysis, and identified gene sets that remained, or became significant. Of the 45 pathways significantly enriched when examining all genes, seven sets remained significant when excluding target genes, and seven new sets became significant, including positive regulation of neurogenesis ($p_{adjusted} = 0.019$, NES = 1.48), regulation of synapse structure or activity ($p_{adjusted} = 0.018$, NES = 1.49), and PI3/AKT signaling ($p_{adjusted} = 0.019$, NES = 1.37) (Supplementary Fig. 3b).

## Electrophysiological features of individual neurons predicted by transcription via NEUROeSTIMator

As an initial application of our model, we examined neurons subjected to direct electrical stimulation. We integrated single cell RNA assays of a Patch-seq dataset[23] with a SMART-seq dataset[18] of mouse GABAergic neurons, both generated by the Allen Institute for Brain Science (AIBS) (Supplementary Fig. 4a). Using the standard Seurat workflow[24], we identified seven clusters corresponding to AIBS-defined neuron types including Sst, Sst Chodl, Pvalb, Vip, Lamp5, Sncg, and Meis2 neuron clusters. We applied NEUROeSTIMator to both datasets and observed elevated activity score predictions in the Patch-seq dataset relative to

the SMART-seq dataset for seven neuron types (Fig. 2a). We obtained empirical p-value estimates of the Kolmogorov–Smirnov (KS) test by permuting expression values in each cell and recalculating activity score for one thousand permutations. Six of the seven neuron types showed significant differences in activity score distributions between datasets (Fig. 2b), of which Lamp5 neurons showed the greatest difference ($p < 0.001$, D = 0.79, KS test) and Vip neurons showing the smallest difference ($p = 0.033$, D = 0.41, KS test). We also noted that several immediate early genes, which serve as predictive targets for NEUROeSTIMator, were among the top genes with higher mean expression in the Patch-seq dataset compared to the SMART-seq dataset (Supplementary Fig. 4b). Together, these results suggest that the Patch-seq protocol induces significantly greater transcriptional upregulation of activity-dependent genes as compared to the standard SMART-seq protocol, and our model is sensitive to these differences. Notably, this effect is consistent for nearly all neuronal types tested.

We established that our model detects transcriptomic signals that separate the electrically stimulated Patch-seq dataset from the unstimulated SMART-seq dataset. However, it is not clear whether elevated activity was due to electrical stimulation and neuron depolarization, or general features of the Patch-seq protocol. Patch-seq neurons showed variability in predicted activity and, because all neurons underwent the stimulation protocol, we hypothesized that variability in activity score predictions were linked to individual neuron differences in electrophysiological features, either intrinsic properties, or responses to stimulation.

To determine if NEUROeSTIMator predictions of activity could be explained by electrophysiology, we fit a lasso-regularized generalized linear model with 10-fold cross validation to predict NEUROeSTIMator output using only electrophysiological features derived from Patch-seq recordings. We used 41 electrophysiological features, including features measured at baseline and in response to the current ramp stimulus. As substantial differences in electrophysiology exist between neuron types, we residualized activity score and derived features for neuron type in order to force the model to learn cell type-agnostic combinations of features that explain variability in activity score. For further analyses, we focus on the four most abundant neuron types: Lamp5, Pvalb, Sst, and Vip neurons. A PCA on these 41 features revealed 90% of variance is explained by fifteen principal components, three of which were associated with activity score ($p < 0.05$) (Supplementary Fig. 4c). The cross-validated lasso model predictions were significantly correlated with NEUROeSTIMator output (R = 0.234, $p < 2.2 \times 10^{-16}$). To evaluate feature importance for each electrophysiological feature, we obtained bootstrap coefficient estimates and standard errors by permuting sample weights for each of 10,000 model fits (Fig. 2c). Three features showed robust non-zero coefficients; membrane time constant (tau) (coefficient = −0.154, standard error = 0.037), input resistance (coefficient: 0.088, standard error = 0.041), and the threshold current to depolarization (coefficient = −0.080, standard error = 0.038). The combination of these features and their direction of effect suggest that the most stimulus-responsive neurons, those which exhibit a transcriptional response, are more excitable than other neurons of the same type (Fig. 2d). We then examined specifically the relationship between input resistance and NEUROeSTIMator predictions in an independent Patch-seq dataset of human excitatory neurons and, again, found a significant positive correlation ($r = 0.497$, $p = 0.011$) (Fig. 2e). Despite the fundamental differences between Patch-seq datasets in species and major neuron class, this successful validation emphasizes the robustness of the relationship between neuronal excitability and transcriptional response to stimulation, as revealed through NEUROeSTIMator predictions.

To better understand how NEUROeSTIMator connects gene expression to electrophysiology, we performed generalized canonical

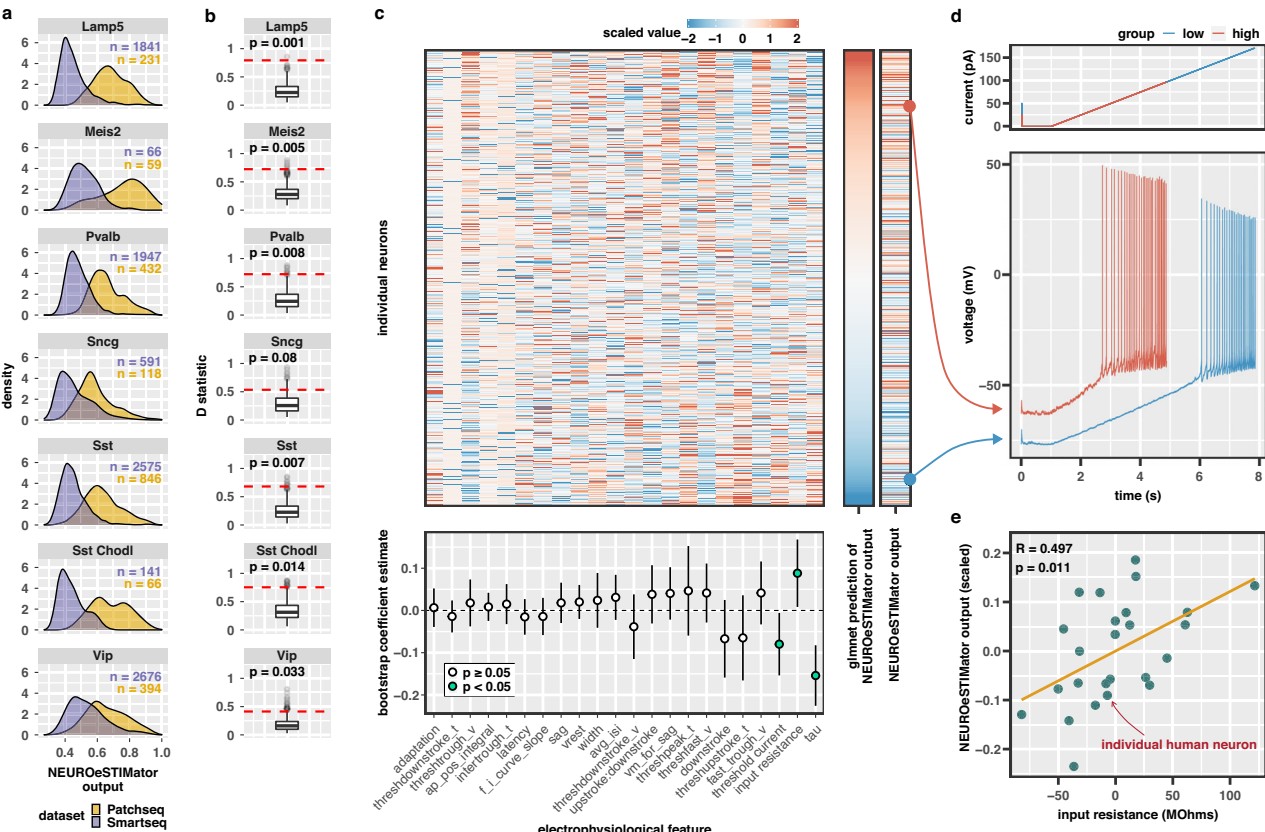

**Fig. 2 | Electrophysiological features of neuronal stimulation captured by transcriptome-based activity score. a** Activity score distributions compared between stimulated Patch-seq neurons (gold) and unstimulated SMART-seq neurons (violet). The number of cells in each comparison are given as text annotations for each density plot. **b** Observed D statistics (red line) and empirical distribution from Kolmogorov–Smirnov tests (two-sided) between activity score distributions of Patch-seq and SMART-seq datasets, using permutations of shuffled expression values ($n = 1000$ permutations per cell type) from all cells represented in **a**. Boxplots depict median value, box bounds denote interquartile range (IQR), whiskers denote values within $\pm 1.5$ x IQR, and points outside of whisker range denote outliers with values $\geq 1.5$ x IQR. **c** Associations between electrophysiological (e-phys) features and predicted activity ($n = 1277$ cells). Heatmap shows feature value columns ordered by the mean bootstrap coefficient, and individual neuron rows ordered by ephys-based lasso model prediction of NEUROeSTIMator output (top). Predicted and observed NEUROeSTIMator output shown to the right. Colors represent z-scaled values, with red colors indicating higher values and blue indicating lower

values. The dot plot (bottom) shows a summary of coefficients estimated by fitting lasso-regularized linear models between electrophysiology feature value and NEUROeSTIMator output ($n = 10,000$ bootstrap permutations per feature). Dots represent the mean coefficient estimate across bootstraps, lines indicate standard error across bootstraps. *P*-values were estimated by counting the number of bootstrap permutations where the coefficient was zero or the opposite sign of coefficients observed from the lasso model fit to all data. Significant features with standard errors that do not cross zero ($p < 0.05$, unadjusted) are depicted by green-colored dots (bottom). See source data for full statistics and feature descriptions. **d** Spike train comparison of two Vip/Ptprt/Pkp2 neurons exhibiting representative electrophysiological features of high (red) and low (blue) NEUROeSTIMator predictions (ramp current injected - top panel, membrane potential - bottom panel). **e** Patch-seq validation of the positive association between input resistance and NEUROeSTIMator predictions in a novel set of human excitatory neurons ($n = 25$ cells). Pearson's correlation test (two-sided) statistics annotated; linear fit (orange line). Source data are provided as a Source Data file.

correlation analysis using three input matrices: the gene expression matrix, a matrix containing the three significant electrophysiology features, and a one column matrix containing the NEUROeSTIMator output. We computed a single component and examined gene expression loadings to understand genes explaining covariance of both predicted activity and the electrophysiological features of excitability. We used GSEA to identify gene sets enriched for the genes correlated to both predicted activity and excitability (Supplementary Data 4). We found significant enrichment of several gene sets including DNA binding transcription activator activity ($p_{adjusted} = 2.74 \times 10^{-5}$, NES = 1.91), MAPK signaling ($p_{adjusted} = 2.54 \times 10^{-4}$, NES = 1.78), and RAS signaling ($p_{adjusted} = 3.80 \times 10^{-2}$, NES = 1.48) (Supplementary Fig. 4d). Genes common to these pathways with high loadings included *Dusp6*, *Dusp10*, *Map3k4*, *Ntrk2*, *Rasgrp1*, and *Prkca*. These pathways constitute an established molecular signaling cascade that functions to link neuron depolarization and related calcium influx to transcriptional response[25,26].

Taken together, these results demonstrate that NEUROeSTIMator captures transcriptomic signatures of activity that are associated with electrophysiological properties measured in response to electrical stimulation of individual neurons. Building upon this successful application in electrophysiology, we next extended our analyses to demonstrate NEUROeSTIMator's utility across a range of applications (Supplementary Data 1).

**Detecting pharmacological activation of neurons**
We applied our model to three single-cell datasets containing neurons from rodents and human cell lines treated with powerful stimulating, pharmacological agents. From the first dataset, we computed activity score for neurons of mice treated with either saline or pentylenetetrazol (PTZ), a depolarizing agent used to model status epilepticus and induce seizures[12,27,28]. As expected, we observed significant differences in distributions of predicted activity (KS test) for 11/14 neuronal subclasses tested, including excitatory cortical layer 2/3 (L2/3) neurons,

Lamp5 interneurons, and Sst interneurons (L2/3 IT CTX: D = 0.40, $p = 9.67 \times 10^{-24}$), (L2/3 IT ENT: D = 0.5, $p = 4.64 \times 10^{-28}$), (Lamp5: D = 0.49, $p = 1.73 \times 10^{-4}$), (Sst: D = 0.42, $p = 3.81 \times 10^{-6}$) (Fig. 3a, Supplementary Data 5).

Next, we compared activity score between neurons from the nucleus accumbens (NAc) of rats treated with either saline or cocaine, a stimulant acting on dopaminergic neurotransmission[29,30]. We found neuron subtype-specific increases in activity score. Two subclasses of dopaminergic medium spiny neurons (MSNs) and both Pvalb interneurons exhibited significant differences in predicted activity (Drd1-MSN: D = 0.15, $p = 5.92 \times 10^{-7}$), (Drd3-MSN: D = 0.19, $p = 0.049$), (Pvalb:

D = 0.22, $p = 0.028$) (Fig. 3b, Supplementary Data 5). Because our model was not trained using any rat or dopaminergic neurons, these findings provide further endorsement for robust estimation of neuronal activity induced by potent pharmacological agents of stimulation.

To further evaluate whether our model could be successfully applied to human data, we examined a dataset of human induced pluripotent stem cell-derived neurons[31]. Neuron cultures were either unstimulated or treated with KCl depolarization[32] buffer for 1, 2, or 4 h. Our model predicted low neuronal activity for the unstimulated group, which was consistent among cell types and biological replicates. Cells treated with KCl for 1 h demonstrated substantial and significant

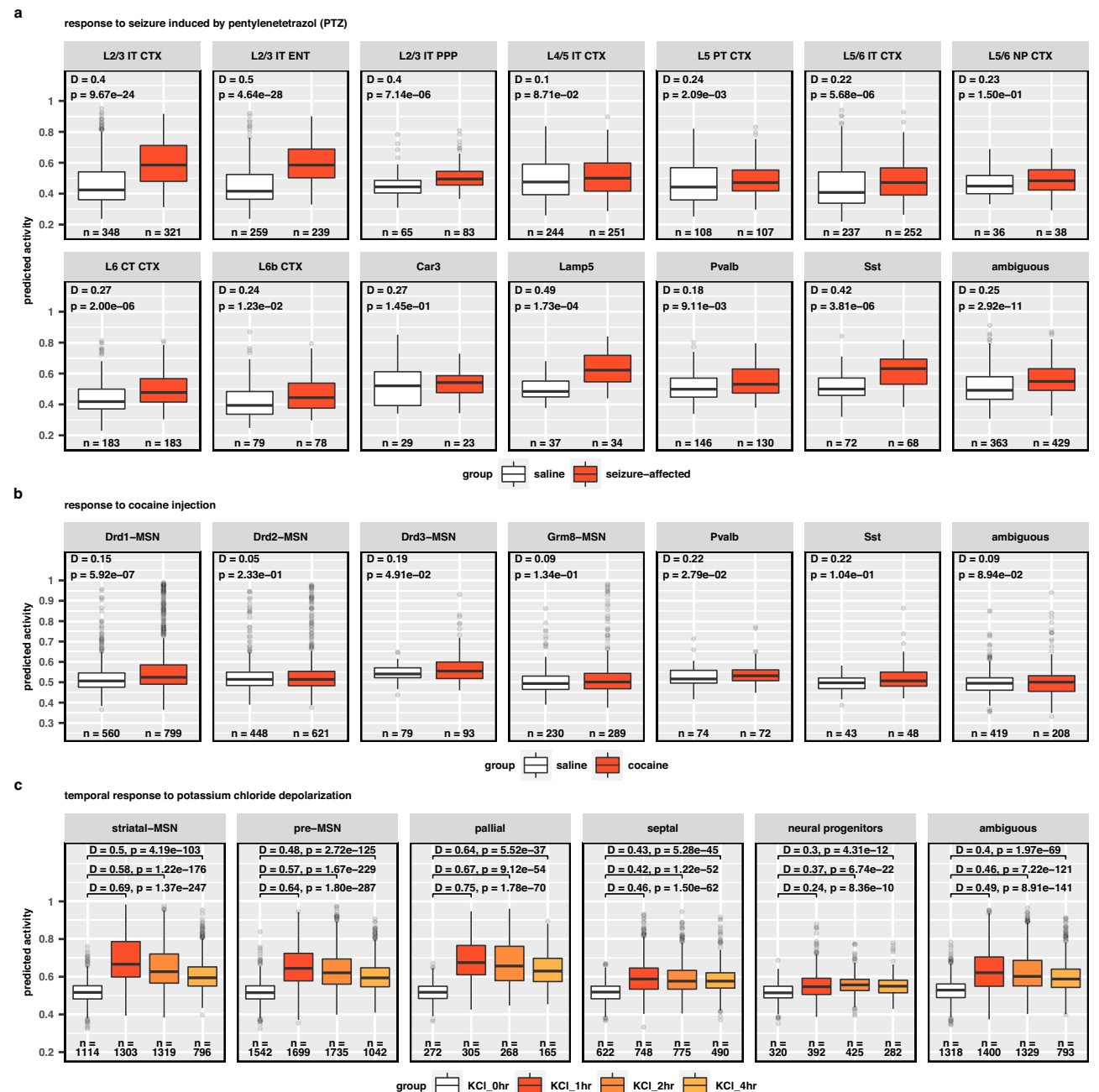

**Fig. 3 | Multi-species generalization of neuronal activity score applied to previously published chemical induction studies.** Predicted activity for various neuron types responding to treatment with potent chemical modulators of neuron activity. **a** Response to PTZ-induced seizure (red) and controls (white) in mouse cortical neurons. **b** Cocaine treatment (red) or controls (white) in rat nucleus accumbens neurons. **c** Time series of predicted activity for human iPSCs treated with depolarizing potassium chloride at 0 h (control), 1 h, 2 h, and 4 h. Annotated statistics from Kolmogorov–Smirnov tests (one-sided) are provided in Supplementary Data 5. Boxplots depict median value, box bounds denote interquartile range (IQR), whiskers denote values within ±1.5 x IQR, and points outside of whisker range denote outliers with values ≥1.5 x IQR. Source data are provided as a Source Data file.

increases in predicted activity. Although significant shifts in predicted activity were observed for all cell types ($p < 0.05$), the activity score of post-mitotic neurons displayed a stronger response to KCl compared to mitotic progenitors. The most responsive neuron types were the mature striatal neurons and pallial neurons (striatal: $D_{1hr} = 0.69$, $D_{2hr} = 0.58$, $D_{4hr} = 0.50$, $p_{1hr/2hr/4hr} < 2.22 \times 10^{-16}$), (pallial: $D_{1hr} = 0.75$, $D_{2hr} = 0.67$, $D_{4hr} = 0.64$, $p_{1hr/2hr/4hr} < 2.22 \times 10^{-16}$) (Fig. 3c, Supplementary Data 5). Notably, all cell clusters except neural progenitors and septal neurons followed a similar temporal pattern of predicted activity modestly declining at 2 h relative to peak activity at 1 h, with a further decline at 4 h (i.e., $D_{1hr} > D_{2hr} > D_{4h}$). Despite the steady decline in predicted activity after 1 h, none of the neuron clusters completely returned to basal activity by 4 h, the final time point in the experiment. The temporal trends of predicted activity are indicative of synaptic depression[33], suggesting NEUROeSTIMator is sensitive to detecting short term plasticity mechanisms.

Together, these analyses suggest our model can robustly assign higher estimates of activity to cells subjected to chemical exposures that are expected to elicit strong and generally ubiquitous transcriptional responses to stimulation.

## Activity score as a generalizable classifier of neuronal stimulation

Next, we asked if our model could detect neuronal activation by more subtle forms of stimuli, such as sensory experience. We applied our model to a dataset containing visual cortex neurons from mice housed in darkness for seven days prior to light exposure for 0, 1, or 4 h[34] (Fig. 4a). We observed a significant increase in predicted activity for neurons from mice exposed to light, relative to controls. To elucidate temporal patterns of activity, we tested differences in activity score distributions between pairs of each time point. Activity score was significantly increased at 1 h for all cell types, in particular for excitatory cortical neurons (layer 2/3: $D = 0.56$, $p < 2.22 \times 10^{-16}$), (layer 4: $D = 0.60$, $p < 2.22 \times 10^{-16}$), (layer 6: $D = 0.54$, $p < 2.22 \times 10^{-16}$). At 4 h of light exposure, predicted activity began to show diverging trends which were foreshadowed by predicted activity at 1 h. Although trending towards a return to baseline, activity scores of neurons at 4 h were not significantly different from neurons at 1 h, while several glial cell types returned to baseline (Fig. 4a, Supplementary Data 6).

As we observed similar trends in temporal activity predictions between the unstimulated (0 h) and 1 h group, we investigated the degree to which the activity score derived from our model could be used as a classifier of experimental group. The degree to which activity score is predictive of experimental group in a particular cell type is expected to represent the robustness of the response in that cell type. Using the visual cortex dataset (VIS) mentioned above, we constructed receiver-operator curve (ROC) plots for neuronal and non-neuronal cell types (Fig. 4b). For both excitatory neuron and interneuron subtypes, the activity score demonstrated varying degrees of predictive power. For example, the activity score alone was able to almost perfectly separate stimulation groups when considering excitatory cortical layer cell types (layer 2/3: AUC = 0.84), (layer 4: AUC = 0.89), (layer 6: AUC = 0.823), as well as certain interneuron cell types (Sst-2: AUC: 0.97), (Vip: AUC = 0.90). Despite the model being trained on purely neuronal cell type populations, the activity score was able to separate stimulation groups for astrocytes (AUC = 0.84) and endothelial cells (AUC = 0.72) with comparable accuracy to neurons.

## Utility in data modalities beyond scRNA-seq

Next, we asked whether our model could identify spatial signatures of learning in brain slices of mice following spatial object recognition (SOR) training - a widely used behavioral paradigm to investigate spatial memory mechanisms[35]. Using spatial transcriptomic data from brain slices of SOR-trained and homecage control (HC) male mice, we applied our model to predict activity for each spot (Supplementary Fig. 5a) and clustered all spots into anatomical regions. The 22 resulting clusters were annotated with brain region naming conventions from the Allen Mouse Brain Atlas (Fig. 5a). In baseline homecage samples, we noted weak activation signatures, primarily covering regions of the striatum and hypothalamus as well as subregions of the hippocampus (Fig. 5b, left). Following SOR training, we noted region-specific increases in predicted activity (Fig. 5b, right). To identify a spatial activation signature of SOR for the entire brain slice, we tested for differences in predicted activity for each brain region cluster (Fig. 5c). We observed significant increases in predicted activity for several cortical and subcortical regions. Multiple layers of the iso-cortex and the retrosplenial area showed increases in activity following SOR (Fig. 5d), with the greatest increases observed in layers 2/3 ($\beta = 0.17$, $p_{adjusted} = 3.16 \times 10^{-4}$) and the retrosplenial cortex ($\beta = 0.17$, $p_{adjusted} = 1.00 \times 10^{-6}$). The caudoputamen area of the dorsal striatum, subregions of the hippocampus, and the olfactory/piriform areas also showed significant increases in activity, of comparable magnitude (caudoputamen: $\beta = 0.12$, $p_{adjusted} = 8.01 \times 10^{-4}$). Subregions of the hippocampus were variably activated following SOR, with the strongest induction observed in the CA1 subregion ($\beta = 0.12$, $p_{adjusted} = 9.14 \times 10^{-5}$), while the dentate gyrus subregion did not show significant activation ($\beta = 0.03$, $p_{adjusted} = 0.07$). Other regions predicted to be least activated by SOR include the thalamus, hypothalamus, and fiber tracts (Supplementary Data 7).

Visium samples contain signal from multiple neuronal and non-neuronal cells. Cortical layers and CA1 showed profound increases in activity but are also known to have relatively high neuron density. Therefore, we considered the extent to which regional neuron density influences coefficient estimates in Fig. 5d. We performed cell type deconvolution on each Visium spot, using neuronal and non-neuronal cell type expression profiles from the Allen Cell Types Database. For a subset of genes with high variability across cell classes, we fit a linear model to each Visium spot as a function of GABAergic, glutamatergic, and non-neuronal expression profiles. Coefficient estimates for each cell class were rescaled across samples to a range of zero to one, and then adjusted for each sample to sum to one, thus providing a proxy measure of cell class density (Supplementary Fig. 6a). We found a significant positive relationship between estimated regional neuron abundance and log-estimated regional effect of learning on activity ($\beta = 0.87$, $p = 0.01$), (Spearman correlation = 0.54, $p = 0.009$) (Supplementary. Fig. 6b). We used the residuals of the linear model in Supplementary Fig. 6b as a neuron density-corrected view of the effect of learning (Supplementary Fig. 6c). Cortical layers, retrosplenial cortex, CA1, and caudoputamen regions show greater learning-induced increases in activity score than expected based on neuron density estimates alone, suggesting that neurons in these regions exhibit a more potent transcriptional response to learning, or that there are other cell types contributing to the activity signature. The dentate gyrus and regions of the hypothalamus, amygdala, thalamus showed lower activity induction than expected.

To validate the finding that the CA1 region was preferentially activated by spatial learning over the dentate gyrus, we performed RNAscope[36] on coronal brain slices from SOR-trained (1 h post-training) and homecage control mice (Supplementary Fig. 5b). We selected three NEUROeSTIMator target genes, *Egr1*, *Egr3*, and *Nr4a1* as probes. For each brain slice, we quantified mean fluorescent intensity (MFI) of each probeand then averaged the MFI of all three probes for each hippocampal subregion. A two-way ANOVA revealed a significant interaction between hippocampal subregions and training groups. In line with the spatial transcriptomic data (Supplementary Fig. 5c), we found that CA1 showed a preferential increase in average probe intensity following spatial learning (Sidak's test, CA1: $p_{adjusted} = 0.0075$, dentate gyrus: $p_{adjusted} = 0.5553$) (Supplementary Fig. 5d).

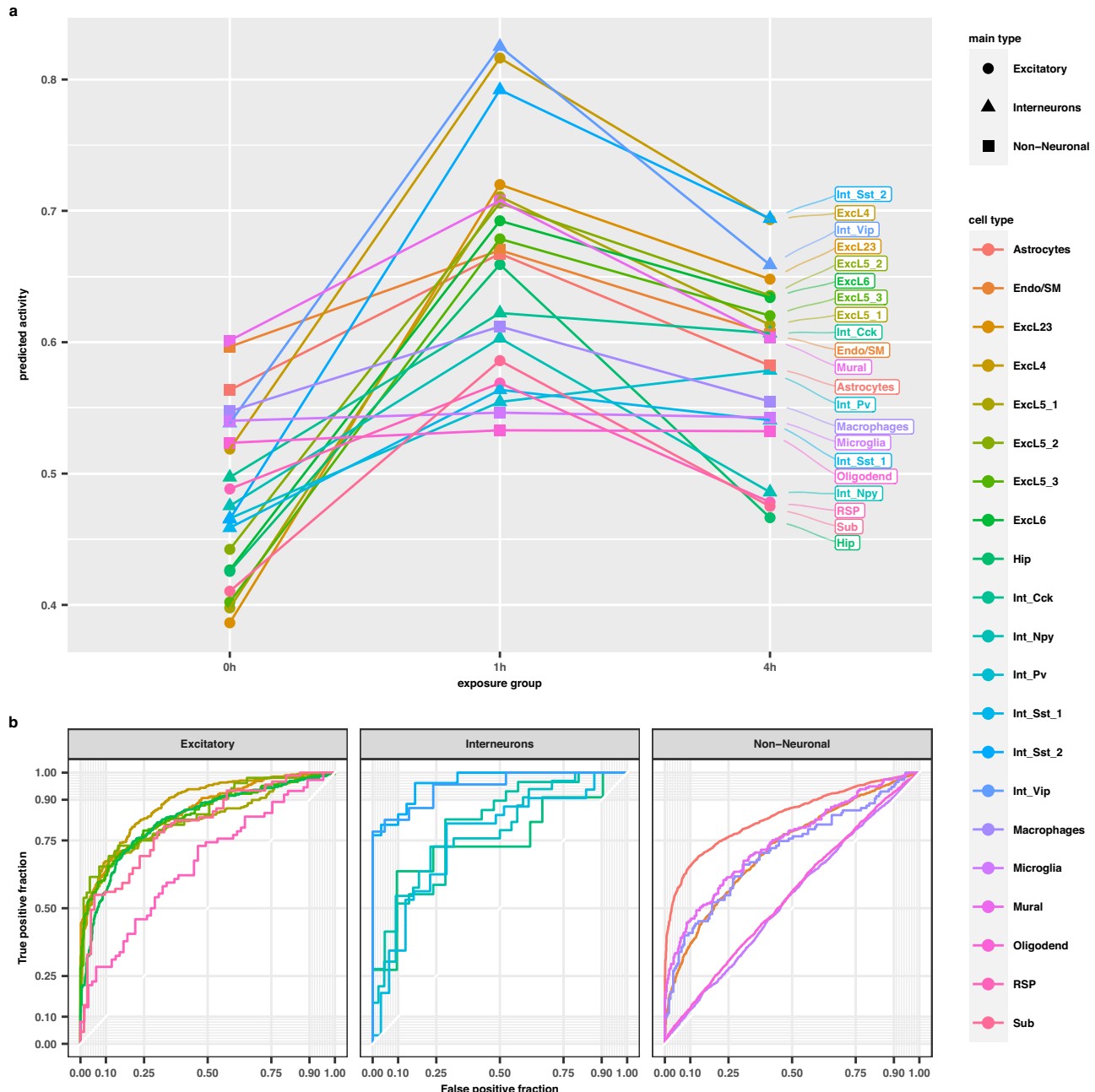

**Fig. 4 | Temporal patterns and classification of in vivo sensory activation. a** Cell type activity predictions of visual cortex neurons in freely behaving mice exposed to light for 0, 1, or 4 h. Activity predictions summarized by cell type and experimental group, and Kolmogorov–Smirnov (one-sided) test results are provided in Supplementary Data 6. Color indicates cell type and shape indicates major cell type groupings. **b** ROC plots indicate the ability of predicted activity to distinguish between the 0 h and 1 h experimental groups. Diagonals from bottom left to top right indicate an accuracy similar to random chance, while lines moving straight vertically, then straight horizontally indicate perfect separation. Source data are provided as a Source Data file.

## NEUROeSTIMator outperforms competing methods in discriminating experimental stimulation groups

The NEUROeSTIMator approach to quantifying activity uses an encoding neural network to infer transcriptome-wide coexpression with a set of 22 activity-dependent target genes. We investigated whether the added complexities of our model are necessary to produce a metric that successfully discerns stimulated samples from unstimulated. We compared NEUROeSTIMator to seven alternative models. The first competing method is a scaled additive model, a simple and easily implementable approach that scales normalized expression of NEUROeSTIMator's 22 target genes across samples to range of [0,1], and then sums scaled values within samples. The second method, the PC1 model, scores samples with target gene loadings derived from the first component of a PCA on the NEUROeSTIMator training dataset. The third method uses MAGIC for graph-based imputation before applying the scaled additive approach to the imputed expression values. The fourth method, the transcriptome-naïve model, is a neural network trained with identical parameters to NEUROeSTIMator, but without access to non-target gene expression inputs. The fifth approach, the transcriptome reconstruction approach, is a neural network trained with identical parameters to NEUROeSTIMator but reconstructs the entire transcriptome with the 22 target genes having equal total loss weight to all other genes. The sixth and seventh approaches, the 2 & 3-unit bottleneck models, are neural networks trained with identical parameters to NEUROeSTIMator, but

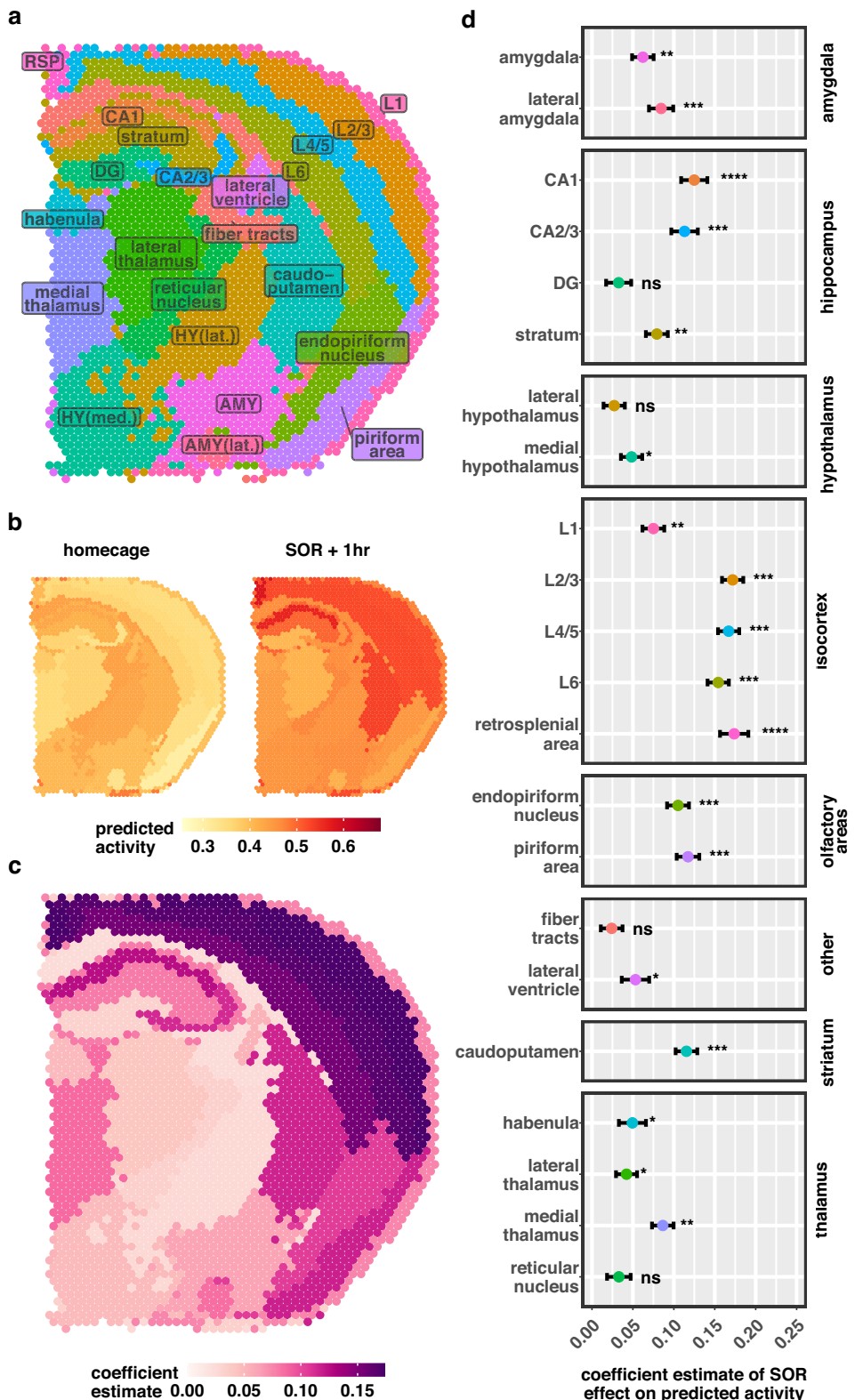

**Fig. 5 | Spatial transcriptomic patterns of neuronal activation after spatial learning. a** Spatial anatomical clustering of RNA-sequencing spots. RSP retrosplenial area, HY hypothalamus, AMY amygdala, med: medial, lat lateral. **b** Activity score per spot, averaged within experimental groups of homecage controls (left) and 1 h after SOR training (right). Yellow color indicates low activity and red indicates high activity. **c** Brain region-specific induction of activity following SOR training. Differences tested using a linear mixed effects model (two-sided), colored by the SOR coefficient estimate per region (*n* = 3 biologically independent mice per group). Dark purple colors represent greater differences between groups.

Cluster-wise differential activity statistics are summarized in (**d**) and provided in Supplementary Data 7. Dots represent estimated coefficients of SOR on activity score; linear fit. Brackets indicate standard error of the coefficient. Asterisks indicate a significant difference in activity score between homecage and SOR groups—false discovery rate (FDR)-adjusted *p*-values are *FDR ≤ 0.05, **FDR ≤ 0.01, ***FDR ≤ 0.001, and ****FDR ≤ 0.0001. Exact *p*-values, FDR, coefficients, and the number of cells per group, per brain region are provided in Supplementary Data 7. Source data are provided as a Source Data file.

using two and three bottleneck units, respectively. We evaluated each model by scoring activity in stimulated and unstimulated samples from six datasets (Supplementary Data 1) and determined separability of the experimental groups using area under the curve (AUC). This approach was applied to datasets featured in Figs. 3, 4, and 5, as well as a novel single nuclei dataset from mouse dorsal hippocampus following SOR. For each method, dataset, and neuron type we computed AUC and summarized the performance differences between NEUROeSTIMator and competing methods (ΔAUC). (Supplementary Figs. 7, 8, Supplementary Data 8). NEUROeSTIMator demonstrated higher performance than each of the competing methods ($AUC_{NEUROeSTIMator}$ = 0.732, $AUC_{scaled.additive}$ = 0.651, $AUC_{MAGIC}$ = 0.587, $AUC_{PC1}$ = 0.661, $AUC_{transcriptome.naive}$ = 0.671, $AUC_{transcriptome.reconstruction}$ = 0.702, $AUC_{NEUROeSTIMator.2units}$ = 0.725, $AUC_{NEUROeSTIMator.3units}$ = 0.717), (mean $ΔAUC_{scaled.additive}$ = +0.081, $ΔAUC_{MAGIC}$ = +0.145, $ΔAUC_{PC1}$ = +0.071, $ΔAUC_{transcriptome.naive}$ = +0.061, $ΔAUC_{transcriptome.reconstruction}$ = +0.030, $ΔAUC_{NEUROeSTIMator.2units}$ = +0.007, $ΔAUC_{NEUROeSTIMator.3units}$ = +0.015). Across all competing methods, datasets, and sample groupings, NEUROeSTIMator increased AUC in 77.9% of 763 comparisons. For comparisons where NEUROeSTIMator outperformed the competing method, the average increase in AUC was +0.077. For the remaining 22.1% of comparisons where NEUROeSTIMator did worse than the competing method, the average decrease in AUC was −0.031. Together, these results suggest that the neural network architecture and NEUROeSTIMator's use of broader transcriptomic signals underlie its enhanced performance compared to alternative approaches.

## Discussion

We introduce NEUROeSTIMator, a versatile tool that estimates neuronal activity from transcriptome-wide single-cell gene expression data. Our method generates an activity score that reflects whole-transcriptome response to stimulation, facilitating the rapid identification and prioritization of neurons displaying stimulus response. NEUROeSTIMator's performance and generalizability are demonstrated through capturing electrophysiological (e-phys) features and detecting activation signatures across various stimulation protocols, neuron subtypes, species, and sequencing technologies, including spatial transcriptomics.

Our analysis reveals that activity-predictive signals are distributed broadly across the transcriptome and are enriched for genes related to external stimulus response, synaptic plasticity, MAP kinase activity, and circadian rhythm. Moreover, we demonstrate a significant link between NEUROeSTIMator's predictions and increased neuronal excitability, utilizing a unique single-cell Patch-seq dataset containing gene expression and e-phys readouts from individual GABAergic murine neurons[23]. These results were further confirmed through a significant association between the activity score and measures of excitability in human excitatory neurons[37].

We showcase NEUROeSTIMator's flexibility and generalizability through secondary analyses of several diverse published datasets, detecting potent pharmacologically induced neuronal activity and generalizing across species and cell types. Importantly, our model also detects physiologically relevant activation in response to sensory experiences and uncovers unexpected activity in non-neuronal cells (e.g., astrocytes), suggesting potential applicability to other cell types and stimulus responses.

In a novel application, we use NEUROeSTIMator to analyze spatial transcriptomic data from mice subjected to a spatial learning task, revealing a brain-wide map of learning-induced activation. This highlights the method's adaptability to spatial transcriptomics (and other varieties of bulk RNA-seq data) and suggests potential for uncovering distinct spatial signatures of transcriptional activation related to cognitive processes.

While our model demonstrates a significant link between transcriptional response and e-phys features in murine interneurons and human excitatory neurons, its applicability to e-phys features in broader cell types is limited by the few available Patch-seq datasets. Additionally, our model does not inherently differentiate stimulation types (e.g., pharmacological vs. naturalistic), and we observed some sensitivity to activity signatures in non-neuronal cell types (a potentially desirable feature, depending on the application). Further, the activity score is limited to detecting activation linked to immediate early gene expression; other potential mechanisms that may be linked to neuronal activation and that act independently of this well-established mechanism may not be detected by our tool. Despite these limitations, NEUROeSTIMator delivers key insights across several neuroscience research domains.

As the first robust and generalizable method for estimating neuronal activation from transcriptomic data, NEUROeSTIMator has broad implications for molecular neuroscience research and challenges the single-gene paradigm of measuring neuronal activity. Our method enables the estimation of cellular activity state, a fundamental variable in gene expression research, paving the way for its integration into single-cell and spatial transcriptomics analyses.

## Methods

### Dataset for model training and evaluation

To train the model, we utilized publicly available datasets provided by the Allen Institute for Brain Science, including a single-cell RNA-sequencing (scRNAseq) dataset of over a million cells isolated from mouse cortical and hippocampal tissue[18], and a single-nuclei RNA-sequencing (snRNAseq) dataset of 76,000 nuclei isolated from human cortical tissue. Hereafter, these datasets will be referred to as the Allen Mouse and Allen Human datasets. Both datasets used the 10X Genomics Chromium system for droplet capture. The Allen Mouse dataset was prepared using the Chromium Next GEM Single Cell 3' v3 reagent kit, while the Allen Human dataset used v2.

### Datasets for model application—publicly available

We downloaded multiple datasets from Gene Expression Omnibus (GEO) to demonstrate the utility of our model. The following GEO accessions were included in analyses: GSE102827[38], GSE106678[13], GSE136656[31], and GSE137763[29].

### Gene identifier mapping

We used the R package biomaRt (version 2.46.3) to map gene identifiers from various annotations used in public datasets, and between species, to a common set of reliably mapped genes[39,40]. Ensembl gene identifiers (Ensembl IDs) were used as the primary identifier for mapping genes, and gene symbols were used as secondary identifiers in cases of ambiguous mapping. The Ensembl release 93 archive (July 2018 release) was used for cross-species gene mapping[41]. Genes with one-to-one orthology between mouse and human, as well as mouse and rat, were selected to facilitate cross species utility. All datasets lacking Ensembl ID annotation contained gene symbols, which were then queried against multiple Ensembl archives to determine which archive maximized identifier mapping rate. For instances, when a gene symbol mapped to multiple Ensembl IDs, identifiers present in the cross-species mapping table were preferentially selected. We provide a helper function for mapping gene identifiers to the feature set used by our model, and we further demonstrate usage in the associated tutorial.

### Choice of neural network target genes

To identify robust markers of neuronal activity for use as targets of the neural network, we intersected lists of stimulus-responsive genes from three published RNA-sequencing experiments. Each publication was from a different group of authors, focused on different brain regions, and used different forms of neuronal stimulation (Supplementary Data 1). One publication categorized stimulus-responsive genes into

three groups−rapid primary response genes (rPRGs), delayed primary response genes (dPRGs), and secondary response genes (SRGs)[21]. As SRGs are thought to demonstrate higher celltype-specificity relative to PRGs[1], only rPRGs and dPRGs were considered from this publication. In another publication, approximately 600 genes upregulated in response to kainic acid treatment in the hippocampus were considered[20].

In a third publication, two sets of KCl-responsive genes were available, one from a brain region enriched in glutamatergic neurons and one enriched for GABAergic neurons[19]. For this study, we sought to intersect the results of both the glutamatergic and GABAergic analyses into one list of genes. Published p-value distributions suggested differences in statistical power between these two analyses, and only statistically significant results were published. To expand the list of genes overlapping between these analyses, we reanalyzed the data using GEO2R to obtain two sets of transcriptome-wide statistics. Using significance rankings from the GEO2R reanalysis, we jointly determined p-value thresholds for each analysis based on rank-rank hypergeometric overlap (RRHO, see Supplementary Fig. 9) and identified genes with p-values below these thresholds in both sets with concordant direction of effect[42]. Because this approach used unconventional p-value thresholding, we additionally required intersecting genes to have an estimated fold change greater than or equal to 0.5 in both analyses.

Intersecting the three lists of stimulus response genes provided 41 candidate target genes. From these 41, four genes were excluded due to ambiguous or failed cross-species gene identifier mapping. We also excluded an additional eight genes that were detected in less than 1% of cells in more than one out of the four species-by-class groups (i.e., mouse-GABA, mouse-glut, human-GABA, human-glut) to mitigate the potential for the model to learn spurious target correlations to cell class markers or species markers. We performed principal component analysis (PCA) on the reaming 29 candidate target genes to assess dimensionality and found that PCs beyond PC1 added very little to the variance explained (Supplementary Fig. 1a). All but one gene had a positive loading on PC1, and several genes had a loading near zero. We identified genes that had extreme positive loadings by comparing them to an outlier threshold for PC1 loadings from a shuffled, null PCA. A scaling factor, the ratio of variance from the observed PC1 to the null PC1, was used to make the loadings comparable. The outlier threshold was defined as the median null loading plus or minus three times the median absolute deviation of the null loadings. We found that 22 of 29 genes had loadings considered to be outliers under the null expectation (Supplementary Fig. 1b). We then did 1000 bootstrap permutations of this analysis, each using random samples of 10,000 cells, and counted the number of permutations where each gene's loading surpassed the outlier threshold. The same 22 genes surpassed the threshold in at least 5% of the permutations, while the other 7 genes did not (Supplementary Fig. 1c). With evidence of robust coexpression, those 22 genes were selected as the final set of targets for our neural network: *Arc, Btg2, Coq10b, Crem, Dusp1, Dusp5, Egr1, Egr3, Fbxo33, Fos, Fosl2, Gadd45g, Gmeb2, Grasp, Junb, Nr4a1, Nr4a2, Nr4a3, Per1, Rgs2, Sertad1,* and *Tiparp.*

### Sample filtering, downsampling, and partitioning

Cells were filtered out if they had less than 5000 total counts, greater than 50,000 total counts, less than 1000 genes detected, more than 10% reads originating from mitochondrial genes, or more than 10% reads originating from the most highly expressed gene. Non-neuronal cells and ultra-rare neuron subclasses with less than 1000 cells were removed, and imbalances among species, neuron type, quality control metrics and naively estimated activity were alleviated using weighted downsampling with hierarchically readjusted weights. Cells were grouped hierarchically by species, class, brain region neighborhood, subclass, and finally, by two-dimensional binning of one source of

technical variation ($\log_{10}$ of total counts), and one source of biological variation (PC1 of target genes). Each group at the leaf nodes of the hierarchy were assigned equal weights summing to one. At each level of the hierarchy, moving from leaves to root, weights were readjusted to sum to one for each group. Using these weights, we downsampled the dataset and calculated Shannon's entropy to measure imbalance at different values of $N$ (Supplementary Fig. 2a) We chose to downsample to 500,000 cells to strike a balance between dataset size and diversity. The effects of downsampling are demonstrated for four subclasses in Supplementary Fig. 2b. The R package groupdata2 (version 1.4.1) was used to create five training folds (91.3%, 456,725 samples) and one test split (8.7%, 43,275 samples). We used the groupdata2 function 'fold' with the parameters 'cat_col' set to the species-class group and 'num_col' set to the target gene PC1 values in order to keep these balanced across folds, but also set the 'id_col' parameter to the subclass label so each subclass was present in some folds but not others. This was done to discourage the undesirable learning of subclass-specific bias in cross-validation.

### Dataset augmentation

The Allen Cell Types Database has higher average total counts per cell compared to many publicly available datasets. To increase representation of lower quality cells, we created augmented samples by synthetically downsampling raw counts using the R package scater[43] (version 1.18.6). For each combination of species and neuron subclass, an equal number of cells were randomly assigned a value of either $10^3$, $10^{3.25}$, or $10^{3.5}$ total counts to be downsampled to. All genes were considered for downsampling.

### Feature normalization and preprocessing

Log-normalization, as implemented in Seurat, was used to normalize input gene expression. Total counts for each cell were calculated by summing only the 10,017 features used by the model. Normalized expression levels for each gene were centered and scaled based on mean and standard deviation estimated from the training data. For cross validation, mean and standard deviation were estimated without the held-out fold.

### Model architecture

The architecture of the model was adapted from DCA[44] (see Supplementary Fig. 10). Briefly, input gene expression is supplied to encoder branches; a series of three fully connected dense layers with ELU activations and batch normalization. ELU activations were chosen over RELU to increase learning speed and performance. The first three hidden layers of the encoders contained 16, 8, and 4 units. The encoder then connects to the information bottleneck, a single-unit dense layer with sigmoid activation. Variations of the single unit bottleneck (i.e., two and three unite bottlenecks) were also investigated. Downstream of the bottleneck, a decoder layer outputs the expression values of the target genes, i.e., the estimated mean parameter $\mu$ of the zero-inflated negative binomial (ZINB) model. A separate parallel encoder branch outputs estimates of the dispersion and dropout parameters. To restrict the bottleneck from capturing biases in the training data, we directly supplied auxiliary inputs (neuron subclass, dataset, and quality control variables) to the decoder. Auxiliary inputs were passed through two hidden layers, producing a six-unit dense layer which was then concatenated with the bottleneck activation as well as the output branch for dispersion and dropout parameters to form penultimate decoder layers. The ZINB loss function was used as implemented in DCA, with the only difference being that our implantation only computes loss for the target genes rather than all inputs to the model. This reconstruction loss is based on the maximum likelihood estimate of the ZINB count model, a common model assumed to underlie the count distributions of single cell RNA-seq datasets. A binary cross entropy loss was added to the bottleneck layer to monitor

classification accuracy. For model applications, the model outputs are not used, but the sigmoidal bottleneck activation value is the metric extracted to index neuronal activity (see Supplementary Fig. 10). Two additional variations of the model architecture were also investigated. One alternative model was restricted to using information from only the 22 target genes. Another alternative model reconstructed two separate output branches, each contributing equally to the total loss: one to reconstruct the 22 target genes, and one to reconstruct the whole transcriptome (i.e., all genes supplied at the inputs).

## Model training

We trained the model using the R package keras, version 2.3.0.0, and Tensorflow version 1.15.0[45]. Training proceeded for 10 epochs using the ADAM optimizer. Gaussian dropout was applied to input expression to simulate uncertainty in measurements. Augmented samples were given the same output as the original data to curtail the learning of depth-dependent information, and augmented samples appeared in the same training batches as the samples they were derived from. We used basic hyperparameter tuning to evaluate the impact of training epochs, learning rate, and regularization parameters such as dropout, L1, and L2 penalty. The final model was selected based on the minimum cross-validation bottleneck layer loss for labeled data (binary cross entropy). Although samples from labeled datasets contributed to the ZINB reconstruction loss, the bottleneck loss did not contribute to weight updates during training to curtail the possibility of information leakage between the training and test splits. Only test split samples were included in any downstream analysis presented in this paper.

## Evaluating feature importance

To evaluate relative importance of each gene on predicted activity, we implemented a strategy adapted from the integrated gradients[22], a popular feature attribution approach used in image classification. Briefly, model gradients were calculated from input genes to bottleneck activation for the test set. For each gene we fit a generalized additive model (GAM) with smoothing between the sample gradients and bottleneck activation. We estimated global feature importance by approximating the integral of the fit. We took the summation of the GAM output for the full range of possible bottleneck activation values [0,1], with a step size of $1 \times 10^{-3}$. Gene set enrichment analysis was run using the R package fgsea (version 1.16.0). We calculated enrichment for gene ontology, Reactome, KEGG, and Wikipathways terms of size between 100–200.

## Comparison of Patch-seq and Smart-seq datasets

The SMART-seq and Patch-seq datasets were integrated, clustered, and visualized using the R package Seurat[24] (version 4.0.1). The sctransform 'v2' method was used for normalization[46]. Cell clustering results were cross-referenced with provided neuron type annotations to identify neuron types common to both datasets and harmonize labels. We used the Kolmogorov–Smirnov test (KS test) to assess distributional differences in predicted activity of samples from the Smart-seq and Patch-seq datasets at the neuron type level. Only neuron types with at least one hundred samples in each dataset were considered. To derive empirical *p*-values and determine whether dataset-specific biases is transcriptome-wide expression levels contributed to observed differences in predicted activity, we ran 1000 permutations of the KS test using activity score predictions from within-sample shuffled gene expression values. Empirical p-values were calculated from the frequency of permutation D statistics that were more extreme than those observed in the actual data.

## Modeling activity score with electrophysiological features

The whole-cell Patch-seq electrophysiology recordings from neurons of mouse visual cortex provided by the Allen Institute for Brain

Sciences were downloaded as NWB files from the DANDI (ID: 000020, September 2021 update)[23]. The dataset included electrophysiological recordings of 4284 specimens from 1040 mouse subjects. The Python package IPFX (version 1.0.5) was used to perform quality control and compute electrophysiological features from the NWB files. We built a generalized linear model with lasso regularization using the R package glmnet (version 4.1-6) to predict output of NEUROeSTIMator using only electrophysiological features. Cells included in the glmnet training belonged to major neuron types (e.g., Vip, Sst) with at least 200 cells, and minor neuron types (e.g., Sst Rxfp1 Prdm8) with at least 10 cells. Features with missing values in five hundred or more cells were removed, and cells with any missing values for the remaining features were discarded. We engineered additional features from the IPFX output, including time and voltage changes between various action potential landmarks (e.g., voltage change between action potential threshold and peak). Minor neuron type was regressed out of the activity score and electrophysiological features. Cells with feature values more extreme than three standard deviations of the mean were considered outliers and removed. Significance of non-zero coefficients from the glmnet model was evaluated with 10,000 bootstrap permutations. In each permutation, a vector of weights is shuffled and used in fitting the glmnet model. For each input feature, we computed the mean bootstrap coefficient. The confidence interval was defined as the mean estimate plus or minus 1.96 times the standard deviation of the bootstrapped coefficients. Feature significance is indicated by the confidence intervals not reaching or overlapping zero. To validate the electrophysiology findings, we did a Pearson's correlation test between NEUROeSTIMator output and input resistance for Patch-seq neurons from the AIBS dataset 'Multimodal Analysis of Human Layer 5 Pyramidal Neurons'[37]. Canonical correlation analysis (CCA) was used to explore gene expression profiles with shared relationships to predicted activity and three electrophysiological features. Inputs to the generalizable CCA included the NEUROeSTIMator output matrix, the matrix of three significant electrophysiological features, and the gene expression matrix. One canonical component was calculated for each input matrix, with the tau parameter set to 0 to maximize correlations between all sets of variables. Gene set enrichment analysis was performed as described above using the gene loadings for the gene expression canonical component.

## Preparation of application datasets

Application datasets were downloaded from the Gene Expression Omnibus (GEO). In datasets without provided labels, cell types were identified by running the standard Seurat pipeline, identifying cell clusters, and cross-referencing cluster marker genes to those used in the source publications or the Allen Cell Types Database. To prepare datasets for input to NEUROeSTIMator, gene expression data was log-normalized sample-wise and scaled gene-wise using mean and standard deviation values from the original training dataset. Samples with at least 1000 counts were then input to the model and activity score was the 1-unit bottleneck activation layer output.

## Testing differences in predicted activity

For all datasets analyzed in Figs. 2 and 3, we used the KS test to evaluate differences in activity score distributions. For statistical tests in Fig. 5, we used the R package lmerTest to fit a linear mixed effect model with donor as a random effect to test for differences in predicted activity of experimental groups across the spatial transcriptomic clusters, deriving p-values and coefficient estimates of learning on predicted activity per cluster. In Supplementary Fig. 6b, we adjusted the coefficient estimates of learning on predicted activity for neuron density. We estimated neuron density using a cell type deconvolution technique. We derived gene signatures for GABAergic neurons, glutamatergic neurons, and non-neuronal cell classes by averaging the median gene

expression for their respective clusters and subclasses from the Allen Cell Types Database.

For each sample in the Visium dataset, we constructed a linear model, correlating the gene expression vector with a three-column matrix that contained class-averaged expression signatures. We then normalized the coefficient estimates across all Visium samples for each cell class by scaling them to a 0–1 range and then adjusting their sum to one within samples. This process provided an estimate of abundance for each cell class. We summed the estimated abundances of gluta-matergic and GABAergic neurons to determine neuron abundance or density for a specific Visium spot, Subsequently, we created a linear model of brain region log coefficient estimates as a function of brain region averaged neuron density estimates. The residuals of this model represent the neuron density-corrected coefficient estimates.

## Performance benchmarking

We compared the performance of NEUROeSTIMator to seven alternative approaches to quantify activity. Performance was defined by the ability to separate labeled cells based on the stimulation status (i.e., experimental group) using area under the curve (AUC). The seven alternative approaches were calculated as follows. The scaled additive measure of activity is calculated by first scaling log-normalized target expression values to a range between 0 and 1, and then summing the scaled targets in each cell or sample. We also applied this scaled additive approach to target expression levels imputed by MAGIC[47], as implemented in the R package Rmagic (version 2.0.3), with default parameters. To calculate the PC1 measure, we first performed PCA on target genes of the training set used to train NEUROeSTIMator. The loadings for PC1 were then applied to the test set. The remaining four approaches were alternative versions of NEUROeSTIMator. The transcriptome-naïve model is a version of NEUROeSTIMator trained with the same parameters, but information from the non-target input genes is blocked by setting their input values and weights to the first hidden layer to zero. The whole-transcriptome model reconstructed the entire transcriptome during training, as opposed to reconstructing only the targets. For this model, the 22 target genes and the broader transcriptome were assigned equal weight in the loss function during training. The final two models were trained with a bottleneck of either two or three units. To summarize these multivariate bottleneck out-puts to a univariate score for AUC calculation, we fit generalized linear models to the binary stimulation group status as a function of the bottleneck outputs in the training data, and then applied the models to test data. To benchmark NEUROeSTIMator against alternative models for binary outcome prediction of stimulation group status, we eval-uated model ability to separate stimulated from unstimulated samples. We computed the Area Under the Receiver Operating Characteristic Curve (AUC-ROC) for each model across all combinations of source dataset and sample groupings. We then determined the AUC differ-ences between NEUROeSTIMator and the alternative models for each sample grouping. To summarize the average AUC difference within each dataset, we calculated the mean of the AUC differences for the respective sample groups. This yielded a single value representing the mean AUC difference for each source dataset in the test set. Lastly, we derived the final performance metric by computing the mean of these mean AUC differences across all data sources, offering a com-prehensive evaluation of the NEUROeSTIMator's performance relative to the alternative models.

## Animals

The spatial transcriptomic and RNAscope experiments were per-formed using male C57BL/6 J mice obtained from Jackson Laboratory (#000664), aged between 2 and 4 months. The mice were housed in groups of up to five individuals per cage, which contained soft bedding material. They had access to food (NIH-31 irradiated modified mouse diet #7913) and water ad libitum. A 12-h light-dark schedule was maintained, with the start of the lights-on period designated as Zeit-geber time zero (ZT 0). Animals were housed in the Animal care facility of the University of Iowa under a pathogen-free condition at the temperature of 21–22 °C and relative humidity of 60–70%. The experimental procedures followed the guidelines for animal care and use set forth by the National Institutes of Health, and they were approved by the Institutional Animal Care and Use Committee (IACUC) at the University of Iowa.

## Behavioral training for spatial transcriptomics

We generated a novel spatial transcriptomic dataset examining the effects of spatial learning in mice. The dataset contains spatial RNA-sequencing of whole brain slices from 1 h after SOR training or home cage controls (HC). 10–12 weeks old C57BL/6 J male mice were trained in SOR task as described previously[35]. Briefly, mice were handled for 5 consecutive days prior to the SOR training. On the day of the training, animals were individually placed in an open field with a spatial cue and was allowed to habituate for 6 min. They were returned to their home cage and the arena was wiped clean with ethanol. The animals were again placed back into the open field with three objects at specific spatial locations. The animals were allowed to freely explore the objects in the arena for 6 min. This training with objects comprised of three trials of 6 min each. One hour after the completion of the train-ing, mice were euthanized and whole brains were flash frozen in iso-pentane placed on dry ice.

## Spatial transcriptomics

Mouse brain section per mouse was cut at 10 μm thickness and mounted onto each Visium slide capture area. After H&E staining, each bright-field image was taken as described in the spatial transcriptomics protocol. Tissue permeabilization was performed for 18 min, as established in the tissue optimization assay. The Visium Spatial Gene Expression Slide & Reagent kit (10x Genomics) was used to generate sequencing libraries for Visium samples. Libraries were constructed according to the 10x Visium library construction protocol and sequenced by Illumina NovaSeq6000. Raw data was then processed using the 10x Genomics Space Ranger analysis pipeline. See Supple-mentary Fig. 5a for images of predicted activity for each replicate.

## In situ hybridization

In situ hybridization was performed on 20 μm coronal brain sections from SOR-trained and HC mice using RNAscope reagents (Advanced Cell Diagnostics). Briefly, fixed frozen brains were sectioned on a cryotome and mounted on Superfrost™ Plus microscope slides (Fisherbrand). Slides then underwent serial dehydration in Ethanol, followed by Hydrogen Peroxide treatment, target retrieval, and Protease III treat-ment—all of which were done according to the manufacturer's protocol. Hybridization of probes was done at 40 °C for 2 h in an HybEZ oven using a 50:1:1 cocktail of *Egr1*, *Nr4a1*, and *Egr3* respectively. The probe signals were amplified with Pre-amplifier 1 (Amp 1-FL) and counter-stained with OPAL dyes (Akoya Biosciences) corresponding to different excitation-emission wavelengths. Finally, the slides were mounted with Vectashield® Antifade Mounting Medium with DAPI (Vector Labora-tories). Slides were stored in 4 °C until they were imaged.

## Confocal imaging and analysis

After performing the RNAscope assay, brain sections were imaged using Olympus FV3000 confocal microscope with a 10X NA = 0.4 objective at 800 × 800-pixel resolution. High magnification images of the hippocampus were obtained using a 40X NA = 1.30 oil immersion objective at 800 × 800-pixel resolution and 1.25 optical zoom. All images (8 bit) were acquired with identical settings for laser power, detector gain and pinhole diameter for each experiment and between experiments. Images from the different channels were stacked and projected at maximum intensity using ImageJ (NIH). Hippocampal

subregion-specific Mean Fluorescence Intensity (MFI) of each RNA-scope probe was evaluated using plugins in ImageJ. The average MFI for all the 3 probes corresponding to each hippocampal subregion was compared between the two experimental groups. Two-way ANOVA was used to assess significance of subregion-by-training interaction ($n = 4$ for each subregion/training combination). A post-hoc Sidak's multiple comparisons test was applied to assess significance of individual subregion differences.

## Reporting summary

Further information on research design is available in the Nature Portfolio Reporting Summary linked to this article.

## Data availability

Spatial RNA-sequencing data, including gene expression measurements, tissue images, spot coordinates, and raw FASTQ files have been deposited in the Gene Expression Omnibus (GEO) repository under the accession code "GSE201610". GEO datasets used in this study can be accessed under the following accession codes "GSE111899", "GSE125068", "GSE55591", "GSE106678", "GSE137763", "GSE136656", "GSE102827". Other datasets used in this study can be accessed through the Allen Institute for Brain Science: "Allen Cell Types Database [https://portal.brain-map.org/atlases-and-data/rnaseq]", "Mouse PatchSeq VIS [https://knowledge.brain-map.org/data/1HEYEW7GMUKWIQW37BO/summary]", "Human PatchSeq L2/3 [https://knowledge.brain-map.org/data/0R94W5U07IHCMVJ4TVK/summary]". Results used in the analyses of this publication have been deposited to Zenodo under the accession https://doi.org/10.5281/zenodo.10183210. Source data are provided with this paper.

## Code availability

NEUROeSTIMator is available at https://research-git.uiowa.edu/michaelson-lab-public/neuroestimator/ as a free R package with installation instructions and a tutorial.

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

## Acknowledgements

This work was supported by NIH grant R01 MH 087463 to T.A., NIH grant R01 DC014489 to J.J.M., NIH grant K99 AG 068306 and the Nellie Ball Trust to S.C., and The University of Iowa Hawkeye Intellectual and Developmental Disabilities Research Center (HAWK-IDDRC) P50 HD103556 to T.A. and Lane Stratharn. T.A. and J.J.M. are also supported by the Roy J. Carver Charitable Trust. We thank the Allen Institute for Brain Sciences for their valuable datasets we used to train our model. We thank the creators and authors of DCA, whose work inspired the approach we implemented in this paper. We also thank Mahesh Shetty for his valuable contributions to discussions about this project. Figure 1 and Supplementary Fig. 10 were created using graphical elements from BioRender.com.

## Author contributions

E.B., S.C., T.A., and J.J.M. conceived the study. E.B. performed data curation, model training, analyses, and visualizations. S.C. and T.A. designed the molecular biology experiments with input from J.J.M. and K.P.G. S.C. performed behavioral experiments, stereotactic surgeries, and single nuclei sequencing. L.C.L. performed Visium spatial transcriptomic sequencing experiments. Y.V. and M.E. processed Visium data. J.R. and M.M. performed RNAscope experiments. U.M. performed statistical analysis and created visualization pertaining to the RNAscope data. E.B. and J.J.M. wrote the article.

## Competing interests

T.A. serves on the Scientific Advisory Board of EmbarkNeuro and is a scientific advisor to Aditum Bio and Radius Health. The other authors declare no conflicting interests.
