## [Peer Review File · Nature Communications]

Using deep learning to quantify neuronal activation from single-cell and spatial transcriptomic dataREVIEWER COMMENTS

Reviewer #1 (Remarks to the Author):

Activity-dependent transcription modulates biological processes that regulate neuron cellular states and behaviors. Its abnormality is linked to various neurological disorders. The methods of estimating neural activity remain pretty scarce for now. In this study, the authors made efforts to fill this gap by proposing a novel computational tool named "NEUROeSTIMator," which employs a deep learning framework to calculate an activity score (0-1) from single-cell measurements (particularly scRNA-seq). While I agree that the problem the authors described is very important and we still lack an excellent computational tool to estimate the neural activation, I found that there are several major issues with the method the authors proposed, which undermines all the innovations and novelty the author claimed in the lengthy discussion section.

Major concerns:

(1) The methodological innovation associated with the method is quite minimal. The authors spent two entire paragraphs (lines 62-87) discussing various conventional dimension reduction methods (e.g., PCA, UMAP) and deep neural network-based dimensionality reduction approaches (i.e., DCA). They further claimed that the DCA-based methods are more flexible in compressing the information and extracting important features. Therefore, the NEUROeSTIMator is built upon the DCA framework activation, particularly for some rare neuron cell populations, or work. I understand that a novel tool/method is often developed based on some fundamental frameworks. Particularly, the auto-encoder framework is very commonly used in the single-cell domain. I have read the manuscripts multiple times to search for their "methodological innovation." In other words, how is their method unique and different from the existing framework? For example, is there a customized loss function or structure that better integrates the biological prior? Unfortunately, I have yet to find a significant improvement over the original DCA framework. The authors did identify and leverage a list of ~32 (or 20) known neural activity biomarkers as the output of the model (reconstruction from the bottleneck single neuron that represents the activity score). However, I wonder if that could be regarded as a significant innovation.

Another significant novelty they claimed is that they compress all the input (thousands of input genes) into a single neuron representing the activation. To me, this is risky as one neuron is limited to representing the entire transcriptome. I understand that the authors want one scalar activity score to represent the activation. However, there are alternative strategies that could circumvent this issue. For example, the authors could use the autoencoder framework to reduce the inputs to 4 or 8 dimensions (latent embedding Z), which will then be used to reconstruct the gene expression for the selected 21 neuron activity biomarkers. Another independent classifier could be added to predict the activity from the Z. Since the model is supervised anyway (neuron cells with and without stimulation), such a classifier can be trained to predict a score in the range (0,1) (with a logistic function) to represent the activity. The feature selection (or biomarker identification) can be performed using sensitivity analysis or the gradient-based approach that the author described. This is just an example alternative strategy that could avoid some potential issues described above, which should at least be explored and discussed in the manuscript to increase the novelty of the methodology.

(2) Another important limitation of the neuronestimateor method is the fact that it relies on ~20 known activity biomarkers. In other words, the model could bias toward known discoveries. As our current understanding of neuron activation, as far as I know, remains limited. Therefore, we may still miss a lot of important biomarkers for neuron activation, particularly for some rare neuron cell populations, or in complex disease patients where the disease might significantly disrupt the neuron activity patterns. In those scenarios, we will need an unbiased approach (or at least we should NOT just focus on those ~20 genes). Alternatively, we could consider giving those known biomarkers higher weights in the reconstruction loss function while other genes will still be considered. The alternative model I described in (1) is also flexible in accommodating the above change (just change the output back to the full input gene vector and revise the loss function accordingly).

(3) There are very limited benchmarking analyses in this study to demonstrate the method's superiority. Yes, the author did compare their method to a simple "additive" model, and their

method indeed shows superior performance. However, the additive method they compared is naive (just a simple sum of normalized gene expression of 20 neuron activation markers). Surprisingly, even such a naive method has comparable performance (at least for early time points), which further raises the necessity to perform more benchmarks with other methods. Suppose the author can not find other methods with similar functions. In that case, the author should at least compare to simple logistic regression or random forest model (predicting the activity score 0/1 from the input vector. For example, the model can be trained with neurons with (1) and without (0) stimulations.

Besides those major comments, the paper also has many minor issues

(1) The author described the deep learning model in all text, which makes it uneasy to follow. A simple schematic overview figure should make it more clear. Also, the author should provide the detailed loss function they employed for their model.

(2) Many figures are missing Panel numbers (e.g., A, B,C), which makes it difficult to relate the figure with its legend.

(3) The authors mentioned that they trained the model for 10 epochs. This is very limited. Is the model converged?

Reviewer #2 (Remarks to the Author):

Overall I find this study not only to be novel, but in the bigger picture to be a step towards unifying high-throughput molecular biology and electrophysiology, so that we can say something about the brain activity consequences of the omics profiling studies of disease, which are becoming increasingly large, important and common. It's from this macro perspective, as much as the specific findings, that I think it is appropriate for publication (after revisions) in Nat Comm. As such, I think that this research is going to have ~2x the interest of the average article, as it's going to pull in two different communities which don't typically talk to each other, as much as would be helpful to. I understand how challenging it is to obtain a suite of experimental results (you had a nice array of pharma, physiological and recent tech applications, which I know didn't happen by accident) in tandem with building computational models. My questions to the authors below largely relate to if the model could easily and substantially be improved by modifying a few relatively arbitrary assumptions, and clarifying all of these points in the text.

Line 116 talks about weighting cells to account for less popular types, but then the next sentence seems to say you literally input an equal number of each cell type into the model, so not sure what was actually done.

What are the genes and functions that predict voltage barrier to depolarization (your main activity features) in patchseq? Right now you are building a predictor of genes which theoretically should be relevant to that, and it's cool that you find they are relevant to a degree, but you could just do some univariate stats to see what biological functions are relevant and might be surprised. Also potential to build a ML model for this purely in patchseq data, May not be enough for your AI model (have you tried?) but other models might work in it directly to establish the target gene set.

How do you propose to interpret data from regions of the mouse brain with more/less microglia or non-neuronal cell types, and will that potentially influence the magnitude or variance in fig 5d?

On the selection of stimulus response genes - triple intersection is reasonable, but to actually prove that would want to see that gene sets only in one or two lists perform worse.

What is the intrinsic dimensionality of the patchseq ephys parameters? Is 90% of the variance on the first PC and so it's reasonable to pull a single value from it, or would a multi-dimensional representation be more appropriate, and could that be wed to more than a single-variable activity value?

The ontological characterization of gene contributing to the activity score doesn't reveal tremendously high p-value. If this diversity is also accompanied by diverse expression profiles, I wonder about the ability to represent them with the current (possibly) overly narrow bottleneck architecture or if modifications might improve performance significantly?

Statement on line 578 " We selected a final set of 20 targets based on consistency of coexpression patterns across datasets and broad cell classes" is not precise enough to know what you did.... maybe modify with "... as follows." if you indeed strictly followed the path of the next few sentences. Indeed, I'd been wondering all along how your 20 genes were distributed across coexpression modules, but then when you talk about picking the "top" genes, but it's not clear what that means in terms of coexpression values.

Also, it seems to me there's a sort of typographic error that prevents me from knowing what you did at a key point. You say onf 583: "...top 20 genes were selected as final the final set of output target genes". I can't tell what's going on from that statement... maybe you had an initial 20 targets and then somehow there's maybe a different 20 at the end? It really matters how these genes were selected, but I can't really even know what questions to ask without this being patched up first. To reiterate, I'm looking for the precise formula for how you ended up with these from your coexpression matrices.

Do all 20 of your final targets fall into a single coexpression module? I can't tell exactly how you picked these, but it seems like your methods are generally pointed towards picking a broadly correlated gene set? If you had genes falling into more than one module do you think your 1D autoencoder and activity measure wouldn't work as well?

Reviewer #3 (Remarks to the Author):

The authors have drafted a manuscript focused on a highly interesting and important topic, namely the extent to which a cell's transcriptome is reflective of its neurophysiological state. The authors utilize a deep learning model adapted from DCA to build a classifier of neuronal activity based on transcriptomic information. Although the proposed model has some appealing features, the data provided are not supportive of the conclusions that NEUROeSTIMator meaningfully outperforms a more traditional additive model of IEG expression, nor that the NEUROeSTIMator activity score is robustly predictive of the neurophysiological properties of individual neurons.

In Figure 2c, the authors report that the "random forest predictions were significantly correlated with NEUROeSTIMator output ($R = 0.11$, $p = 7.86 \times 10^{-10}$)". NEUROeSTIMator output explains only ~1.2% of the variance in the random forest predictions, for which the highly significant p-value is likely driven by the large sample size.

In Figure 2d, the authors tested for differences in voltage barrier between the high and low activity groups in four types of neurons. However, these are group-level differences, rather than predictions at the level of individual neurons. Moreover, these group-level differences are between "high" versus "low" activity groups, which leaves out predictions for the substantial proportion of neurons that are found in the intermediate range of activity scores between "high" and "low" activity.

In the analyses of pharmacological activation shown in Figure 3, these are again group-level analyses rather than for individual neurons. Moreover, in none of these datasets is electrophysiological data available. Rather, the authors infer a broad concept of "neuronal activation" on the basis of biological knowledge of the pharmacological agents, which greatly reduces the utility of these data to the extent that these analyses are being used to validate the ability of the NEUROeSTIMator to predict electrophysiological activity at the level of individual neurons.

In addition, for the KCl and visual cortex (light stimulation) datasets, the longitudinal time series would be a unique opportunity to examine the accuracy of the NEUROeSTIMator under well

controlled conditions. However these datasets do not contain electrophysiological data to validate whether the NEUROeSTIMator activity scores are indeed predictive of neurophysiological activity at the level of individual neurons.

Lastly, in comparing the performance of the NEUROeSTIMator against an additive model for group-level discrimination, the authors conclude that the NEUROeSTIMator has “drastically” and “substantially” improved performance, yet over half of the cell types examined have a delta AUC < 0.05 and $\sim 15\%$ of cell types have an AUC less than 0.

NCOMMS-22-48085

Using Deep Learning to Estimate Neuronal Activation from Single-Cell and Spatial Transcriptomic Data

RESPONSE TO REVIEWERS: SUMMARY

We are grateful to the editor and reviewers for their enthusiasm and valuable feedback on our initial submission. We have addressed every reviewer comment, which we feel has substantially improved the rigor of our analyses and the overall quality of our manuscript. Notably, we have expanded the performance benchmarking and have included new data related to electrophysiology results, which has further strengthened our findings and conclusions. After consideration of reviewer comments, we updated the model architecture and shifted our training approach from purely unsupervised to semi-supervised learning. Further, we have eliminated design decisions that the reviewers considered arbitrary and clarified others with additional supplemental analyses. We anticipate that our findings and tool will be of great interest especially to the neuroscience research community, and we hope that the editor and reviewers will find the revised manuscript suitable for publication.

Reviewer #1:

Remarks to Authors

Activity-dependent transcription modulates biological processes that regulate neuron cellular states and behaviors. Its abnormality is linked to various neurological disorders. The methods of estimating neural activity remain pretty scarce for now. In this study, the authors made efforts to fill this gap by proposing a novel computational tool named "NEUROeSTIMator," which employs a deep learning framework to calculate an activity score (0-1) from single-cell measurements (particularly scRNA-seq). While I agree that the problem the authors described is very important and we still lack an excellent computational tool to estimate the neural activation, I found that there are several major issues with the method the authors proposed, which undermines all the innovations and novelty the author claimed in the lengthy discussion section.

Major concerns:

1.0 *The methodological innovation associated with the method is quite minimal. The authors spent two entire paragraphs (lines 62-87) discussing various conventional dimension reduction methods (e.g., PCA, UMAP) and deep neural network-based dimensionality reduction approaches (i.e., DCA). They further claimed that the DCA-based methods are more flexible in compressing the information and extracting important features. Therefore, the NEUROeSTIMator is built upon the DCA framework, particularly for some rare neuron cell populations, or overwork. I understand that a novel tool/method is often developed based on some fundamental frameworks. Particularly, the auto-encoder framework is very commonly used in the single-cell domain. I have read the manuscripts multiple times to search for their "methodological innovation." In other words, how is their method unique and different from the existing framework? For example, is there a customized loss function or structure that better integrates the biological prior? Unfortunately, I have yet to find a significant improvement over the original DCA framework. The authors did identify and leverage a list of ~32 (or 20) known neural activity biomarkers as the output of the model (reconstruction from the bottleneck single neuron that represents the activity score). However, I wonder if that could be regarded as a significant innovation.*

Response: We thank the reviewer for their appreciation of the importance of this area of research. We have updated the text to succinctly state the methodological innovations in a clear way.

The methodological innovation of the work we describe is fourfold: 1) the vast, integrated dataset of hundreds of thousands of neurons from multiple data sources, with sophisticated downsampling to ensure even representation of cell types 2) the careful and data-driven selection of immediate early genes (IEGs) to serve as model targets 3) the application of deep learning to this problem and 4) architecture designed to protect the latent space from capturing cell type biases. These and other technical innovations have allowed us, for the first time, to estimate gene expression indicators of cell activation that are consistent with electrophysiological features of increased excitability. Although no such tool existed prior to this manuscript, in this revision we have made good faith efforts to compare NEUROeSTIMator's performance to a variety of approaches, which we hope will satisfy the reviewers regarding technical innovation. With these additional details, we have endeavored to satisfy the reviewer's request for attention to brevity in the discussion. More detail on methodological innovation is provided below.

1) The dataset. To train the model we used hundreds of thousands of neurons from the large-scale and diverse Allen Cell Types Database. We also used multiple labeled (with respect to experimental group) datasets containing stimulated and unstimulated neurons, as well as synthetically augmented cells.

2) **Data-driven curation of activity marker genes.** Many studies rely on a small number of markers to identify activated neurons (e.g., staining, *fos*-based genetic approaches). We utilized differential gene expression lists from three diverse studies of neuronal activity to identify a set of approximately 30 marker genes. We then identified a subset of 22 marker genes that robustly load onto a single principal component, thus providing us a coexpression module containing 22 activity marker genes to base our model on.

3) **Applying deep learning approaches to the problem of neuron activity estimation.** Deep learning is rapidly becoming a powerful tool for genomic and transcriptomic analysis, and autoencoders, in particular, are popular in the arena of single cell transcriptomics, especially for clustering. However, we are not aware of any other method that has attempted to model transcriptomic signatures of neuron activity, neither with current techniques such as neural networks, nor in the context of single cell RNA-seq.

4) **Protecting the latent space from capturing cell type biases.** We provide cellular metadata to the model to relieve the latent space from pressure to learn a representation biased by celltype of dataset. As an example, if 25% of type A neurons show high expression of activity markers but only 10% of type B, a biased representation could learn that type A *cell type* markers (rather than true activity markers) are associated with IEG expression. This scenario leads to different cell types occupying different ranges of the bottleneck space (e.g., type A: 0.5-1, type B: 0-0.5). This could be particularly problematic for a low-capacity bottleneck. To overcome this, we supplied cell type, dataset, and quality control information to an auxiliary decoder branch of the architecture downstream of the latent space, to absolve the latent space from the loss pressure to find cell type or dataset-specific intercepts (see **Supplemental Figure 10**). This causes different cell types to all approximate our imposed bottleneck distribution instead of occupying different ranges of it. Importantly, since this auxiliary metadata information used in the training process is input downstream of the latent space, it does not impact the latent space and is therefore not required to use NEUROeSTIMator.

To our knowledge, nothing like what we describe here has been done before. DCA is not a comparable method, as it is a model-building tool, and does not utilize any of the innovations listed above. It was merely the framework we based our architecture upon.

1.1 Another significant novelty they claimed is that they compress all the input (thousands of input genes) into a single neuron representing the activation. To me, this is risky as one neuron is limited to representing the entire transcriptome. I understand that the authors want one scalar activity score to represent the activation. However, there are alternative strategies that could circumvent this issue. For example, the authors could use the autoencoder framework to reduce the inputs to 4 or 8 dimensions (latent embedding Z), which will then be used to reconstruct the gene expression for the selected 21 neuron activity biomarkers. Another independent classifier could be added to predict the activity from the Z. Since the model is supervised anyway (neuron cells with and without stimulation), such a classifier can be trained to predict a score in the range (0,1) (with a logistic function) to represent the activity. The feature selection (or biomarker identification) can be performed using sensitivity analysis or the gradient-based approach that the author described. This is just an example alternative strategy that could avoid some potential issues described above, which should at least be explored and discussed in the manuscript to increase the novelty of the methodology.

Response: The single-unit bottleneck in our model is only learning a representation for the 22 stimulus-responsive target genes, not the entire transcriptome, and is the appropriate choice for our model for several reasons, which we have further elucidated in the manuscript thanks to the reviewer's input (see **Supplementary Figure 1**, Results lines 86-97, and Methods lines 484-530).

As the reviewer points out, we do want a single scalar value to represent activity for the sake of simplicity and interpretability. A single score is easier to understand, visualize, compare across samples, and to integrate into downstream analyses such as classification or regression (e.g., as a covariate). While a bottleneck with additional capacity would undoubtedly capture more complex patterns, it would also become more challenging to interpret the relationships between multiple dimensions, thus adversely impacting the tool's ease of use in the intended applications. Further, the powerful regularizing effect of reducing bottleneck capacity forces the model to focus on only the most salient information and discard less relevant information. This makes models more robust to noise and reduces the risk of overfitting. Finally, in our revision we show that the final 22 genes we selected as model targets were the subset of genes that robustly loaded onto the first principal component of a PCA performed on a larger set of 29 candidate genes (see **Supplemental Figure 1c**). In other words, the PCA supported the hypothesis that these genes represent a single coexpression module and can be readily captured with our model's single-unit bottleneck.

We are appreciative of the reviewer's suggestions here because they inspired us to shift our model towards a semi-supervised learning approach (in the initial submission, we trained on unlabeled data in a form of unsupervised feature learning). We describe the new approach on lines 571-606. In brief, we included partitions of the various application datasets described throughout the manuscript into the training process (for monitoring of training progress and model selection; see details below). Just like cells from the main Allen Institute dataset, the model training learns to reconstruct the expression of cells from these other labeled datasets. To add supervision to the training process, we attached an additional output connected directly to the 1-unit bottleneck that predicts labels for the labeled datasets only. Though the labeled samples from the training set were not involved in any downstream analyses featured in the manuscript, we were cautious about the potential for information leakage from the training set to the test set because they were from the same labeled datasets. For this reason, we chose to remove this output's contribution to the training loss, and instead used it to monitor whether the autoencoder reconstruction loss was actually promoting a learned bottleneck representation that helps discriminate labels. Although it did not directly impact weight updates during training, we used the validation loss from this new output to select a final model from cross-validation. This semi-supervised approach yielded the updated model we present in our revision.

1.2 Another important limitation of the neuroestimator method is the fact that it relies on ~20 known activity biomarkers. In other words, the model could bias toward known discoveries. As our current understanding of neuron activation, as far as I know, remains limited. Therefore, we may still miss a lot of important biomarkers for neuron activation, particularly for some rare neuron cell populations, or in complex disease patients where the disease might significantly disrupt the neuron activity patterns. In those scenarios, we will need an unbiased approach (or at least we should NOT just focus on those ~20 genes). Alternatively, we could consider giving those known biomarkers higher weights in the reconstruction loss function while other genes will still be considered. The alternative model I described in (1) is also flexible in accommodating the above change (just change the output back to the full input gene vector and revise the loss function accordingly).

Response: We thank the reviewer for this observation, and we have updated the text throughout the manuscript (see lines 42-45, 62, and 103-105 for examples) to emphasize two key points. First, the model utilizes the whole transcriptome (not just 22 genes) to de-noise signal contained in the expression of the 22 immediate early genes (IEGs) used as targets. Second, we emphasize the decades-long history of research on immediate early genes (IEGs) as markers of neuronal activity. While we agree that in general it is better to assume a weak prior on the state of our knowledge regarding marker genes, in the case of IEGs, there is little reason to suspect that there are unknown first-line gene expression pathways that have evaded detection over decades of research across thousands of labs – it is difficult to think of another class of marker genes that could more aptly be called a “gold standard.” Our careful testing and curation

of the IEG target genes (see **Supplemental Figures 1 and 9**) is another example of the methodological innovation the reviewer requested.

1.3 There are very limited benchmarking analyses in this study to demonstrate the method's superiority. Yes, the author did compare their method to a simple "additive" model, and their method indeed shows superior performance. However, the additive method they compared is naive (just a simple sum of normalized gene expression of 20 neuron activation markers). Surprisingly, even such a naive method has comparable performance (at least for early time points), which further raises the necessity to perform more benchmarks with other methods. Suppose the author can not find other methods with similar functions. In that case, the author should at least compare to simple logistic regression or random forest model (predicting the activity score 0/1 from the input vector. For example, the model can be trained with neurons with (1) and without (0) stimulations.

Response: We thank the reviewer for this feedback. In addition to the scaled additive approach from our initial submission, we have added two additional methods to benchmark NEUROeSTIMator performance. We summarize the findings of our expanded benchmarking analysis in **Supplementary Figure 7**, where we depict the difference in AUC between NEUROeSTIMator and each of the comparisons for various groups of labeled data from the test set. The mean AUC of the “naive” scaled additive approach across all cell groupings was 0.541. One of the new comparisons, which we refer to as the ‘PC1’ approach, uses target gene principal component loadings from the training data to compute PC1 scores for each cell in the test set. Not only does this approach have the advantage of being derived from hundreds of thousands of neurons, but it also serves as a baseline dimensionality reduction comparison that lacks non-linearities and broader transcriptomic inputs. This approach outperformed the scaled additive approach with a mean AUC of 0.658. The other new comparison method we added was to train a version of NEUROeSTIMator with identical parameters but cut off all information from non-target inputs. We refer to this as the ‘targets only neural network’, or ‘targets only NN’ for short. We implemented this by initializing the same model architecture NEUROeSTIMator was built from, and then set all non-target inputs to zero as well as freezing their weights and bias to zero. We confirmed these frozen weights were not updated during training. This method contains all the non-linearities and regularization of NEUROeSTIMator but cannot utilize information from non-target genes in the broader transcriptome. This approach marginally outperformed the ‘targets PC1’ method with a mean AUC of 0.666. NEUROeSTIMator outperformed all other comparisons with a mean AUC of 0.724. We would point out that in the machine learning literature generally, such an improvement in AUC over the previous state of the art would be seen as noteworthy or even appreciable (see <https://www.nature.com/articles/s41467-020-20657-4>). More details can be found in the methods on line 701, results on line 368, and the data is depicted in **Supplementary Figure 7 and 8**.

1.4 Besides those major comments, the paper also has many minor issues

(1)The author described the deep learning model in all text, which makes it uneasy to follow. A simple schematic overview figure should make it more clear. Also, the author should provide the detailed loss function they employed for their model.

Response: We thank the reviewer for this recommendation. In our revision, we have provided a diagram of the model architecture (see **Supplementary Figure 10**). Our ZINB reconstruction loss function is the same as the one described in DCA, and we have added additional information about it on lines 578-589, and we have made the loss function code accessible on our GitLab repository.

1.5 (2) Many figures are missing Panel numbers (e.g., A, B,C), which makes it difficult to relate the figure with its legend.

Response: We apologize for this oversight and thank the reviewer for catching it. We have updated the figures and legends to be easier to follow in the revision.

1.6 (3)The authors mentioned that they trained the model for 10 epochs. This is very limited. Is the model converged?

Response: Due to specifics of the model architecture and the size of the training data set, training convergence was indeed rapid. Specifically, we used ELU activations on the dense layers of the encoder and decoder, batch normalization after each of those dense layers, and used ADAM as the optimizer. We also tested multiple epoch and learning rate parameters and ultimately selected the final model based on cross-validation (again, with model selection being guided in part by the loss observed in the labeled data).

Reviewer #2:

Remarks to Authors

Overall I find this study not only to be novel, but in the bigger picture to be a step towards unifying high-throughput molecular biology and electrophysiology, so that we can say something about the brain activity consequences of the omics profiling studies of disease, which are becoming increasingly large, important and common. It's from this macro perspective, as much as the specific findings, that I think it is appropriate for publication (after revisions) in Nat Comm. As such, I think that this research is going to have ~2x the interest of the average article, as it's going to pull in two different communities which don't typically talk to each other, as much as would be helpful to. I understand how challenging it is to obtain a suite of experimental results (you had a nice array of pharma, physiological and recent tech applications, which I know didn't happen by accident) in tandem with building computational models. My questions to the authors below largely relate to if the model could easily and substantially be improved by modifying a few relatively arbitrary assumptions, and clarifying all of these points in the text.

2.0 Line 116 talks about weighting cells to account for less popular types, but then the next sentence seems to say you literally input an equal number of each cell type into the model, so not sure what was actually done.

Response: We thank the reviewer for their enthusiasm for our work and, in particular, for this comment. We have clarified the description of sample weighting in the manuscript on lines 537-549. The full original dataset was highly imbalanced in terms of species, class label, subclass label, QC metrics, and target gene expression. For example, roughly 25% of the entire dataset consisted of mouse layer 4 cortical neurons. We downsampled the dataset to increase relative diversity of all neuron populations, as well as to improve training efficiency. To accomplish this resampling, we gave each cell a weight inversely proportional to its frequency in the original dataset. For example, in each neuron subclass, we set the sum of weights for cells expressing target genes to be equal to the sum of weights for cells with no target gene expression. Because there were fewer cells expressing target genes than not, each target-expressing cell had a higher weight. We did this in a nested manner, adjusting these weights so that the sum of weights for each neuron subclass was equal. We similarly adjusted the total weights for GABAergic cells to equal to the sum of weights for glutamatergic cells. If a specific neuron subclass only has one cell with relatively high target gene expression, this approach virtually guarantees it will be retained in sampling. We tried downsampling to several choices of N and monitored Shannon's entropy, a measure of class imbalance, at each value of N, for all levels of cell type grouping (species, cell class, brain region neighborhood, cell subclass, and a grouping based on quantiles of sequencing depth and PC1 values for target gene expression). We ultimately chose 500,000 to strike a balance between dataset diversity and training efficiency. We have included a new figure (see **Supplemental Figure 2**) showing the entropy metrics as well as the preferential retention of rare neuron populations.

2.1 What are the genes and functions that predict voltage barrier to depolarization (your main activity features) in patchseq? Right now you are building a predictor of genes which theoretically should be relevant to that, and it's cool that you find they are relevant to a degree, but you could just do some univariate stats to see what biological functions are relevant and might be surprised. Also potential to build a ML model for this purely in patchseq data, May not be enough for your AI model (have you tried?) but other models might work in it directly to establish the target gene set.

Response: This is an excellent point, and we thank the reviewer for this comment, which inspired a new analysis (see description of CCA below). It's an interesting idea to build a model directly on the Patch-seq data. One could predict the electrophysiology features alone to establish the target genes or predict them

alongside the target genes used for NEUROeSTIMator to guide the training process towards learning a mutually informative representation. The current challenges are the scarcity of Patch-seq datasets and the lack of ideal controls (i.e., cells undergoing the Patch-seq protocol without injected current). However, in the spirit of this idea, we used canonical correlation analysis (CCA) to explore gene expression profiles with shared relationships to both predicted activity and the electrophysiology features. We then performed gene set enrichment on the CCA loadings and found several significant gene ontology terms related to general cell response systems. Those include MAPK signaling and regulation, transcription activator activity, RAS signaling, and NTRK signaling. Some of the genes common to these pathways with the highest loadings included *Dusp6*, *Dusp10*, *Map3k4*, *Ntrk2*, *Rasgrp1*, and *Prkca*. Further details can be found on lines 208-222 and **Supplemental Figure 4d**.

2.2 How do you propose to interpret data from regions of the mouse brain with more/less microglia or non-neuronal cell types, and will that potentially influence the magnitude or variance in fig 5d?

Response: We thank the reviewer for another excellent idea, which inspired another analysis and figure (see lines 334-354 and **Supplemental Figure 6**). The reviewer brings up a great point – each measurement sample from the Visium data is composed of multiple brain cells of different types. As we have demonstrated, the model is not necessarily specific to neuron activity, so our predictions on any sample with properties of bulk tissue may represent a combination of activity signatures from multiple cell types. Furthermore, in samples with varying neuron densities, coefficient estimates for differential activation may be amplified or diluted depending on which cell types have been activated and the percentage of cells responding. We performed a new cell type deconvolution analysis on our spatial transcriptomic data to investigate whether neuron density influences our learning-induced coefficient estimates in **Figure 5d**. We confirm that, indeed, the coefficient estimate is correlated with estimated neuron density (Spearman correlation = 0.52, $p = 0.01$). To interpret the coefficient estimates, we propose to first residualize them for the region-averaged estimate of neuron density. Depending on the question of interest, one might be interested in either the raw coefficients, or the neuron-density-corrected coefficients.

2.3 On the selection of stimulus response genes - triple intersection is reasonable, but to actually prove that would want to see that gene sets only in one or two lists perform worse.

Response: Our new PCA analysis (see **Supplemental Figure 1**) provides additional rigor and clarity to the selection of target genes and addresses the reviewer's comment here. Further, we have included a brief analysis examining the average feature attributions of non-target genes that were present in 0, 1, 2, or 3 of the lists and show that, on average, the more lists a gene was present in (of 3), the greater the gene's contribution to model (see **Supplemental Figure 3**). This analysis demonstrates a core methodological innovation of NEUROeSTIMator – that genes throughout the transcriptome can provide meaningful information to the model's predictions, without being a target gene itself, through associations with target gene expression.

2.4 What is the intrinsic dimensionality of the patchseq ephys parameters? Is 90% of the variance on the first PC and so it's reasonable to pull a single value from it, or would a multi-dimensional representation be more appropriate, and could that be wed to more than a single-variable activity value?

Response: The reviewer brings up an excellent point, and the analyses below demonstrate the aspects of electrophysiology that NEUROeSTIMator does and does not capture (it most robustly captures neuron excitability). There are other latent dimensions to the e-phys features that are neither correlated with

NEUROeSTIMator nor other latent gene expression signatures (this may include confounding technical variables that are beyond our reach). This is a highly interesting point, but we would point out that 1) such an exercise is beyond the scope of the current work and 2) it would require a Patch-seq dataset orders of magnitude beyond what is currently possible (our non-Patch-seq training data set here was 500,000 cells, vs. the ~4,000 available Patch-seq interneurons). To the reviewer's specific point, fifteen principal components were required to capture approximately 90% of the variance in the e-phys features of the Patch-seq cells; three PCs were significantly correlated with our activity score (see also **Supplementary Fig. 4c**).

2.5 The ontological characterization of gene contributing to the activity score doesn't reveal tremendously high p-value. If this diversity is also accompanied by diverse expression profiles, I wonder about the ability to represent them with the current (possibly) overly narrow bottleneck architecture or if modifications might improve performance significantly?

Statement on line 578 " We selected a final set of 20 targets based on consistency of coexpression patterns across datasets and broad cell classes" is not precise enough to know what you did.... maybe modify with "... as follows." if you indeed strictly followed the path of the next few sentences. Indeed, I'd been wondering all along how your 20 genes were distributed across coexpression modules, but then when you talk about picking the "top" genes, but it's not clear what that means in terms of coexpression values.

Also, it seems to me there's a sort of typographic error that prevents me from knowing what you did at a key point. You say onf 583: "...top 20 genes were selected as final the final set of output target genes". I can't tell what's going on from that statement... maybe you had an initial 20 targets and then somehow there's maybe a different 20 at the end? It really matters how these genes were selected, but I can't really even know what questions to ask without this being patched up first. To reiterate, I'm looking for the precise formula for how you ended up with these from your coexpression matrices.

Do all 20 of your final targets fall into a single coexpression module? I can't tell exactly how you picked these, but it seems like your methods are generally pointed towards picking a broadly correlated gene set? If you had genes falling into more than one module do you think your 1D autoencoder and activity measure wouldn't work as well?

Response: We thank the reviewer for the invitation to develop a more rigorous approach for selection of the target genes. In our revision, we revisited this early decision point and used a more reasoned approach to selecting target genes. Ultimately, this led us to a slightly different set of target genes and thus required us to rebuild the training dataset and rerun cross validation. The ultimate performance and broad conclusions, however, remain comparable to our initial submission.

Intersecting the three lists of stimulus response genes provided 41 candidate target genes. Four genes were excluded due to ambiguous or failed cross-species gene identifier mapping. We excluded an additional eight genes that had poor detection rates (detected in less than 1% of cells for any of the four species-by-class groups, i.e., mouse-GABA, mouse-glut, human-GABA, human-glut). We did this to mitigate the potential for the model to learn spurious correlations to cell type markers or species.

At this point we considered the possibility that the remaining 29 candidate target genes could comprise multiple coexpression modules. If our targets fell into multiple coexpression modules, the model would struggle to recapitulate them with a 1-unit bottleneck. For reasons outlined above in response to reviewer one, we felt the 1-unit bottleneck was the most appropriate architecture choice, so we needed to select a core set of target genes that belong to a single module. We did not find compelling evidence supporting multiple coexpression modules, but we did find that some of the 29 genes were simply not correlated with any of the other candidates. WGCNA, a common coexpression module-detecting approach, was only able to find one module, though we did not include this analysis as we could not get a good scale-free topology fit (too few genes) as outlined in the WGCNA documentation, which could invalidate the result.

We only mention it here because it is something we investigated. Next, we performed a principal component analysis (PCA) on the candidate target genes and found that PC1 explained a majority of the variance (~15%; 3x greater than PC2) and would be the only component selected per the scree or elbow method heuristic (**Supplemental Figure 1a**). Examining the gene loadings onto PC1, we found that all but one gene had a positive loading, though several genes had a loading near zero. We identified genes that had ‘extreme’ loadings by comparing them to an outlier threshold based on PC1 loadings from a shuffled, null PCA. The outlier threshold was defined as the median null loading plus or minus three times the median absolute deviation of the null loadings. A scaling factor, the ratio of variance from the observed PC1 to the null PC1, was used to make the loadings comparable. We found that 22 of 29 genes had loadings considered to be outliers under the null expectation (**Supplemental Figure 1b**). We then did 1000 bootstrap permutations of this analysis, each using random samples of 10,000 cells, and counted the number of permutations where each gene’s loading surpassed the outlier threshold. Seven genes produced PC1 loadings that exceeded the null expectation in less than 5% of the bootstrap samples and were thus discarded from the target set because of their lack of consistency with the dominating IEG expression pattern (i.e., PC1, see **Supplemental Figure 1c**). With evidence of robust coexpression, we selected those 22 genes as the final set of targets for our neural network. Details of this analysis were added to the methods on line 485.

Reviewer #3:

Remarks to Authors

The authors have drafted a manuscript focused on a highly interesting and important topic, namely the extent to which a cell's transcriptome is reflective of its neurophysiological state. The authors utilize a deep learning model adapted from DCA to build a classifier of neuronal activity based on transcriptomic information. Although the proposed model has some appealing features, the data provided are not supportive of the conclusions that NEUROeSTIMator meaningfully outperforms a more traditional additive model of IEG expression, nor that the NEUROeSTIMator activity score is robustly predictive of the neurophysiological properties of individual neurons.

3.0 In Figure 2c, the authors report that the “random forest predictions were significantly correlated with NEUROeSTIMator output ($R = 0.11$, $p = 7.86 \times 10^{-10}$)”. NEUROeSTIMator output explains only $\sim 1.2\%$ of the variance in the random forest predictions, for which the highly significant p -value is likely driven by the large sample size.

Response: We thank the reviewer for the enthusiasm for this area and our initial submission. With respect to this point, although the correlation between the NEUROeSTIMator output and the random forest predictions was relatively weak, it was highly significant. In our revision we approached this analysis using a glmnet model instead of a random forest to simplify the interpretability of electrophysiological feature importance. In the revised analysis, the cross-validated glmnet reaches a considerably higher correlation of $r = 0.234$.

3.1 In Figure 2d, the authors tested for differences in voltage barrier between the high and low activity groups in four types of neurons. However, these are group-level differences, rather than predictions at the level of individual neurons. Moreover, these group-level differences are between “high” versus “low” activity groups, which leaves out predictions for the substantial proportion of neurons that are found in the intermediate range of activity scores between “high” and “low” activity.

Response: We appreciate the reviewer's comment and would like to make clear that all predictions are indeed at the level of individual neurons, both in the initial submission and in this revision. In **Figures 2, 3, and 4**, and especially in panel **2a,c,e**, note that individual NEUROeSTIMator predictions are for individual cells (largely neurons). The only NEUROeSTIMator predictions in this manuscript that are not at the individual cell level are those for the spatial transcriptomics data, which are for individual spots. We have updated the manuscript to make clearer to the reader that NEUROeSTIMator is designed to work at the single cell level (though as demonstrated it has utility in bulk gene expression settings). Further, in the revised analysis we have included the full range of predictions of individual neurons in the Patch-seq dataset. In addition, we now more extensively describe the connection between electrophysiological features and NEUROeSTIMator output (see **Figure 2** and **Supplemental Figure 4**). In brief, we find evidence that the composite gene expression signature captured by NEUROeSTIMator predicts excitability in individual Patch-seq murine interneurons. This was further validated in a second Patch-seq data set (new in this revision) of human excitatory neurons, demonstrating the generalization of this signal across major neuron types and species. We feel that our extensively revised analysis and addition of new data on this specific point provides a compelling response to the reviewer's comment.

3.2 In the analyses of pharmacological activation shown in Figure 3, these are again group-level analyses rather than for individual neurons. Moreover, in none of these datasets is electrophysiological

data available. Rather, the authors infer a broad concept of “neuronal activation” on the basis of biological knowledge of the pharmacological agents, which greatly reduces the utility of these data to the extent that these analyses are being used to validate the ability of the NEUROeSTIMator to predict electrophysiological activity at the level of individual neurons.

Response: We appreciate the reviewer’s concerns, and we would like to clarify the intent and scope of our model: to identify single-cell transcriptional signatures of stimulus response that generalize across various neuron types and stimulation paradigms, rather than to predict precise electrophysiological states of neurons. Inferring a “broad concept of neuronal activation” is exactly the intent of our model. Consequently, neither the analyses of pharmacological agents, nor any application from **Figures 3, 4, or 5** were intended to validate the Patch-seq findings. It’s actually the other way around: the fact that we observed robust association with e-phys indicators of excitability was itself a compelling validation of our ability to infer a “broad concept of neuronal activation” from gene expression.

Furthermore, the Allen Institute’s Patch-seq dataset is unique in throughput, cell type representation, and accessibility. To our knowledge, it is currently the only publicly available dataset of its kind. While we believe a model capable of robustly inferring electrophysiological measures of activity from single-cell RNA-seq data would be an incredible advancement of neuroscience, this option is currently precluded by the scarcity of Patch-seq data. However, despite these limitations, we did include an analysis of a new Patch-seq dataset of human neurons from the Allen Institute in our revised manuscript, which corroborated our initial observations in the mouse interneurons. On lines 199-206, we use this dataset to validate our finding that input resistance of a neuron is predictive of NEUROeSTIMator output (see **Figure 2e**).

The group-level analyses featured throughout the manuscript demonstrate that NEUROeSTIMator can discern between labeled stimulation groups. In every dataset we analyze except for the Patch-seq data, the stimulus is not administered to cells individually. Drugs are administered to bulk tissue or cultures, or complex brain circuits are activated through natural means such as sensory exposure or learning. The true neuron activity state of individual neurons is not known in these data. In these experiments, the ground truth of stimulated vs unstimulated exists only at the group level, and so consequently we did group-level analyses. Still, each observation in each group is, in fact, a prediction on an individual neuron.

3.3 In addition, for the KCl and visual cortex (light stimulation) datasets, the longitudinal time series would be a unique opportunity to examine the accuracy of the NEUROeSTIMator under well controlled conditions. However these datasets do not contain electrophysiological data to validate whether the NEUROeSTIMator activity scores are indeed predictive of neurophysiological activity at the level of individual neurons.

Response: As the KCl and light stimulation datasets lack e-phys features, they cannot validate our electrophysiology findings, although that was not their intended purpose. In our revised manuscript we have added an abundance of clarification regarding the purpose of each application dataset.

3.4 Lastly, in comparing the performance of the NEUROeSTIMator against an additive model for group-level discrimination, the authors conclude that the NEUROeSTIMator has “drastically” and “substantially” improved performance, yet over half of the cell types examined have a delta AUC <0.05 and ~15% of cell types have an AUC less than 0.

Response: In the revision, we have edited the text to have a more measured and humble tone. Still, NEUROeSTIMator offers a measurable and meaningful improvement over the other options that we benchmarked against (mean AUC of 0.724; AUC improvements over other methods ranging from 0.058 to 0.183). Across all competing methods, NEUROeSTIMator has a delta AUC > 0 in 88.1% of all cell groupings tested. On average, NEUROeSTIMator increased AUC by +0.110 while, in the remaining 11.9% of groupings where NEUROeSTIMator was outperformed, the average decrease was -0.031, which is less than a third of the magnitude of NEUROeSTIMator's average improvement (see lines 368-400 and **Supplemental Figure 7**). As mentioned above in our response to reviewer 1, such improvements in AUC would be seen as substantial in the machine learning literature (see for example <https://www.nature.com/articles/s41467-020-20657-4>); our AUC improvement is at least double the AUC improvement shown in this previously published example.

REVIEWER COMMENTS

Reviewer #1 (Remarks to the Author):

I sincerely appreciate the authors' diligent efforts in revising the manuscript and addressing prior comments. They have made considerable progress, resolving several major concerns. However, despite the commendable effort, I maintain key reservations that prevent me from endorsing the publication of this manuscript in its current form.

1. My central concern lies in the authors' continued reliance on approximately 20 biomarker genes to quantify neural activity. Their diligent curation of these established biomarkers is praiseworthy, and the selected genes may indeed be the best currently available set. Yet, I still firmly believe that using the complete transcriptome can provide a richer, more nuanced picture of neuronal activity for several reasons:

Completeness: The full transcriptome offers a comprehensive view of all the genes expressed at a particular moment. Such an approach doesn't limit itself to genes directly involved in neural activity but also those indirectly affected or associated with related processes.

Sensitivity: Alterations in neuronal activity might not consistently result in changes in the expression of the selected biomarker genes, yet it could impact other genes. The entire transcriptome analysis permits the detection of such subtler shifts.

Robustness: A focus on a few biomarker genes could leave the results susceptible to noise or variable expression levels of these genes. The entire transcriptome analysis spreads this risk across many more genes, potentially boosting the robustness of results.

Complexity: Neuronal activity isn't a simple binary phenomenon. Its complex nature, with varied types and patterns, could lead to different gene expression patterns. The use of the entire transcriptome enables capturing this intricacy.

Potential for Novel Findings: Utilizing the entire transcriptome can lead to the discovery of new genes or pathways involved in neural activity, which could provide fresh insights into neuronal function.

I acknowledge the authors' attempts to utilize the entire transcriptome for denoising the data and enhancing the signal of the ~20 genes. However, I suggest that the authors further investigate this approach by comparing it to other existing denoising methods such as MAGIC, which could provide a deeper understanding of the proposed method's performance.

Finally, while I acknowledge the authors' diligent curation of the ~20 biomarkers, solely basing the quantification of neural activity on these genes may limit the potential for novel discoveries and introduce a bias towards existing knowledge.

Instead, I urge the authors to integrate the existing knowledge into their deep learning network as regulatory parameters (or other strategy that could leverage the prior) to empower unbiased and more systematically neural activity quantification with deep neural network. This will not only present more comprehensive and complete neural activity quantification, it will also enable the discovery of potential novel biomarkers for neural activity.

2. The authors have responded comprehensively to my comments regarding novelty. However, I perceive potential issues that need further attention:

Definition of Innovation: The response could benefit from a more precise articulation of what defines 'innovation' in this context.

Claim of Uniqueness: The authors assert that no other method has attempted this type of modelling. Although this might be true, providing more specific references or evidence could substantiate this claim.

Data Collection: The authors emphasize the importance of their dataset's size and diversity. While data collection is critical to any study, it's not typically considered a methodological innovation.

Deep Learning Application: The authors mention using deep learning but could explain more thoroughly how they have applied it innovatively.

Auxiliary Decoder Branch: A clearer explanation, possibly accompanied by a figure or diagram, could help clarify this aspect of their methodological innovation.

In conclusion, while I appreciate the work that has gone into this manuscript, I believe that further revisions addressing these concerns (particularly the 1st major comment) are needed to sufficiently elevate its scientific contribution. I look forward to seeing how this work evolves and advances our understanding of neural activity.

Reviewer #2 (Remarks to the Author):

I appreciate the authors' detailed responses, which are not mere verbiage, but accompanied by time-consuming and thoughtful analysis and fairly deep reconfiguration of their model. In particular I admire your CCA, additional test of the triple intersection for selecting genes, and comments on the impact of glial proportions were slick. You've clearly got a diversely-talented yet integrated team working on this.

There is one essential remaining point that's been brought up repeatedly, that has not been reasonably addressed. That is the issue of a single dimension bottleneck. The rationale for this is summarized by the authors below, with my comment in brackets. Generally I thought this justification lacked the data support and high rigor of all other responses.

A single score is easier to understand [actually reviewers are having a harder time understanding it because it is so limited as to be difficult to square with ephys], visualize [millions of informative 2d plots published each year], compare across samples [we routinely compare many dimensions across samples and, anticipating your response, while data reduction is helpful, I don't think anyone will balk at a couple of variables], and to integrate into downstream analyses such as classification or regression (e.g., as a covariate) [actually it looks straightforward]. While a bottleneck with additional capacity would undoubtedly capture more complex patterns [we don't know if they will be complex or not], it would also become more challenging to interpret the relationships between multiple dimensions [you don't know what without testing, a frankly anything is more challenging than no comparison (1D), but could also supply information that is useful], thus adversely impacting the tool's ease of use in the intended applications [we live in a world of 100k-variable datasets popping out every day, so going from a 1d to 2d model cannot realistically be an objection].

I would suggest that exploring 2 and 3-node inner layers of your encoder is actually not contrary to your current position - if you try that and find it is not useful for various reasons, then you have actual support for the current model. If that's the case, in the future paper, you just have a sentence like: We explored a large bottleneck but found problems xyz. However, if you do make it work, you're opening the door to a more general methodology for inter-relating multi-scale systems vs more of a denoising mechanism (I'm simplifying, but I think you get it).

This question is entwined with the findings that it takes 15 PC's to hit 90% of ephys variance and PC1 only providing 15% of the variance of your gene set. That latter is portrayed as high, but actually it's quite low for a set of genes thought to be functionally related. Both of these point to the potential for adding a few nodes to the bottleneck. I hope you can see from my prior questioning, I think this work has great potential, and I'm not just asking for stuff as an exercise, but think it's fundamental to realizing the direction on which you've already embarked.

Regards,
Chris Gaiteri

Reviewer #3 (Remarks to the Author):

The authors have made substantial improvements to the manuscript, in both the revisions to the text and additional analyses. Although the variance explained still remains small using a glmnet model ($\sim 5.4\%$) and the AUC improvements are somewhat modest, the results and models presented do represent an important step forward in understanding the relationship between the cellular transcriptome and its neuronal activity. I expect this manuscript to be well cited, and look forward to its reception and impact on the field upon publication.

RESPONSE TO REVIEWER COMMENTS

Reviewer #1 (Remarks to the Author):

REVIEWER: I sincerely appreciate the authors' diligent efforts in revising the manuscript and addressing prior comments. They have made considerable progress, resolving several major concerns. However, despite the commendable effort, I maintain key reservations that prevent me from endorsing the publication of this manuscript in its current form.

1. My central concern lies in the authors' continued reliance on approximately 20 biomarker genes to quantify neural activity. Their diligent curation of these established biomarkers is praiseworthy, and the selected genes may indeed be the best currently available set. Yet, I still firmly believe that using the complete transcriptome can provide a richer, more nuanced picture of neuronal activity for several reasons:

RESPONSE: We thank the reviewer for these comments and for their commitment to the rigor of this manuscript. We have done as the reviewer asked, and in brief, the results indicate that including the transcriptome as a predictive target does not improve performance. The reviewer's comments and philosophy on this subject seem intuitive, however, the results of our investigations at the reviewer's behest suggest that the resulting latent representation learned by the model is inferior for this task. We detail below the investigations we undertook and their results. These results suggest that reconstructing the whole transcriptome is counterproductive when the goal is a latent signal that predicts the immediate early gene response linked to neuronal activation. We hope that the reviewer is satisfied by our extensive good faith efforts to investigate this hypothesis.

REVIEWER: Completeness: The full transcriptome offers a comprehensive view of all the genes expressed at a particular moment. Such an approach doesn't limit itself to genes directly involved in neural activity but also those indirectly affected or associated with related processes.

RESPONSE: We agree in principle with the reviewer, though we would argue that completeness is already achieved by virtue of the inclusion of the whole transcriptome at the inputs. During training, signal from the whole transcriptome is consolidated (e.g., elicited in part by dropout layers, which tend to distribute predictive signal from the inputs) to best predict genes well-established to be linked to neuronal activity. This approach is consistent with the general observation in machine learning at large that noise is better tolerated at the inputs vs. in the training labels. The 22 target genes, which are directly involved in neuronal activity, serve to dictate the representation the model learns. However, we must emphasize that those representations are in fact informed by the whole transcriptome. We demonstrate in **Supplemental Figure S3 (see lines 124-131)** that several other non-target genes are impacting the latent space (i.e., activity score), and therefore are impacting the reconstruction of the 22 target genes. These are the genes that, as the reviewer states, are indirectly affected by

neuronal activity or associated with related processes, and our approach is able to leverage information from them because it has a comprehensive view of the transcriptome, which is directly input to the model. Consequently, we see this as compelling evidence that the current model architecture achieves the “completeness” that the reviewer mentions. On the other hand, as our results below demonstrate, forcing the model to reconstruct (e.g., at the output) the whole transcriptome adds significant noise to the task and hinders its accomplishment.

REVIEWER: Sensitivity: Alterations in neuronal activity might not consistently result in changes in the expression of the selected biomarker genes, yet it could impact other genes. The entire transcriptome analysis permits the detection of such subtler shifts.

RESPONSE: Our target gene selection criteria identified targets that are consistently affected by neuronal activity in multiple studies. Furthermore, there is ample evidence in the literature characterizing our target genes as being induced by neuronal activity. We are not aware of any other module of activity-dependent genes that are truly independent of these genes. If such expression modules exist, our model will not be able to pick up on that signal. We acknowledge that this is a limitation of our model, though the recommendation to reconstruct the entire transcriptome does not solve this, as demonstrated in our new results (**lines 399-404**). The target genes we selected constitute our prior belief, based on exhaustive examination of the literature, of which genes are responsive to neuronal activity. If activity induces changes to expression that overlaps, or is associated with the induction of our target genes, we can find that signal just by using the transcriptome as an input. If activity induces changes that have evaded all previous attempts to identify them, then we simply cannot model that type of transcriptional activation without any prior information. Finally, it is perhaps helpful to reiterate the primary purpose of this tool, which is to robustly identify cells and tissues that show evidence of neuronal activation. Discovering wholly new mechanisms in the way the reviewer describes, while possible (and even demonstrated in this work, e.g., in **Supplemental Figure 3b**), is not the principal aim of this work.

REVIEWER: Robustness: A focus on a few biomarker genes could leave the results susceptible to noise or variable expression levels of these genes. The entire transcriptome analysis spreads this risk across many more genes, potentially boosting the robustness of results.

RESPONSE: The reviewer makes an excellent point, that focusing on a small number of genes leaves results susceptible to noise. In fact, this was one of our core motivations for this project. In the broader neuroscience community, it is incredibly common to use expression of a single gene (typically *Fos*) as an indicator of neuron activity (see Hudson 2018⁷, Kawashima 2014⁸, Guenther 2013⁹, and Liu 2014¹⁰). We have built a model that uses prior information of activity-dependent changes to examine neuronal activity through the lens of 22 genes rather than just one. Further, our whole-transcriptome input provides supporting information from an even larger pool of genes coexpressed with our 22 activity-dependent genes, and this seems to precisely satisfy the comment this reviewer has made here. In our previous revision, we demonstrated that the addition of the whole-transcriptome *at the input* does indeed make

the model more robust in comparison to model with only the 22 targets as input (see the 'Transcriptome Naive' comparison in our benchmarking analysis, Supplementary Figures 7-8, lines 399-404). Conversely, forcing the reconstruction of the whole transcriptome at the output dilutes the focus of the learning task, effectively inducing noise in the bottleneck.

REVIEWER: Complexity: Neuronal activity isn't a simple binary phenomenon. Its complex nature, with varied types and patterns, could lead to different gene expression patterns. The use of the entire transcriptome enables capturing this intricacy.

RESPONSE: We agree that neuronal activity is not a simple binary phenomenon. Although there are complex patterns of activity, we have relatively simplistic models of molecular kinetics of the resulting transcriptional activation that results from the neuronal activity. The transcriptional response to activity is structured in waves, where an initial wave of immediate early gene transcription and translation gives rise to a downstream wave of effector gene transcription. Most of the complexity in transcriptional response to neuron activity exists in the downstream waves, as they have been shown to vary by neuron type. Our model reconstructs expression of immediate early genes, meaning its predictions pertain to initial waves of transcriptional response. This initial wave, with its transient nature, can be modeled as a binary phenomenon. We acknowledge that inability of our model to identify the highly variable and cell type dependent downstream waves of transcription is a limitation. However, the model can only learn representations for what is present in the training data, and there are only two datasets containing time points where complex downstream transcription is expected, and they are from different brain regions, different cell types, and different stimuli. We need far more data from later time points in different datasets to begin considering a model that could extract this information in a robust way. Ultimately, any attempt to incorporate highly complex and variable transcription patterns goes directly against our goal to create a broadly generalizable model to quantify neuronal activity. We believe that models of the complex downstream transcriptional waves would be useful to the neuroscience field, but we view such an endeavor as a future direction and outside the scope of this paper. In response to this comment, we have expanded our discussion of limitations in the manuscript (see lines 454-456).

REVIEWER: Potential for Novel Findings: Utilizing the entire transcriptome can lead to the discovery of new genes or pathways involved in neural activity, which could provide fresh insights into neuronal function.

RESPONSE: Since we used the whole transcriptome as an input to our current model, the model was able to learn associations with our target genes and genes in the broader transcriptome. In **Supplemental Figure 3 (lines 131-145)**, we examine the functions and pathway annotations of informative genes. We emphasize that we did look at the non-target genes (genes that we did not have prior expectation for being involved in neuronal activity). As an example of a novel finding, we found that informative non-target genes were even more enriched for neurogenesis and PI3K/AKT signaling functions than the target genes.

REVIEWER: I acknowledge the authors' attempts to utilize the entire transcriptome for

denoising the data and enhancing the signal of the ~20 genes. However, I suggest that the authors further investigate this approach by comparing it to other existing denoising methods such as MAGIC, which could provide a deeper understanding of the proposed method's performance.

RESPONSE: We thank the reviewer for this suggestion. In this revision, we have added MAGIC as a competing method for comparison to our model (see **Supplemental Figure 7-8, lines 730-732 & 399-404**). MAGIC returns an imputed matrix of gene expression values, so we apply the scaled additive approach to the imputed matrix to produce a single score for comparison to NEUROeSTIMator. The models in our benchmarking analysis were trained with a training set, and the AUC comparisons were done for the test set to ensure a fair comparison with fresh data not seen by any of the models. We were unable to identify functionality in MAGIC that would allow a training set model to be built, and then subsequently applied to a test set. We tried running MAGIC with training and test set combined, as well as the test set alone, and found similar benchmarking performance. Consequently, because of these limitations, the performance of the MAGIC approach is, if anything, an overestimate. Ultimately, we found that MAGIC was the lowest performing competing model (**Supplemental Figure 7**). Furthermore, a crucial advantage NEUROeSTIMator has over MAGIC is that it can be applied 'out-of-the-box' and does not need to be retrained or recomputed on each dataset it is applied to.

REVIEWER: Finally, while I acknowledge the authors' diligent curation of the ~20 biomarkers, solely basing the quantification of neural activity on these genes may limit the potential for novel discoveries and introduce a bias towards existing knowledge. Instead, I urge the authors to integrate the existing knowledge into their deep learning network as regulatory parameters (or other strategy that could leverage the prior) to empower unbiased and more systematically neural activity quantification with deep neural network. This will not only present more comprehensive and complete neural activity quantification, it will also enable the discovery of potential novel biomarkers for neural activity.

RESPONSE: In response to these comments, we have created a new model, described on **lines 386-389, 613-615 & 738-740**, that reconstructs expression of both 1) the set of 22 target genes as well as 2) the broader transcriptome. We leveraged the prior information by splitting these gene sets into two separate outputs, and weighted them equally in the loss function. We trained the model and added the resulting latent space to our performance benchmarking analysis. We found that our current NEUROeSTIMator model (i.e., without whole transcriptome reconstruction) outperforms the whole transcriptome output model with a mean AUC delta of 0.03 (see **Supplemental Figure 7, lines 399-404**). This decrease in ability to differentiate stimulated from unstimulated cells suggests that the addition of the transcriptome output is detracting from the signal represented in the latent space of our current model.

2. REVIEWER: The authors have responded comprehensively to my comments regarding novelty. However, I perceive potential issues that need further attention:

Definition of Innovation: The response could benefit from a more precise articulation of what defines 'innovation' in this context.

RESPONSE: We thank the reviewer for their invitation to state the innovation of this work more clearly. Building on the NIH's framework of innovation, we view innovation as not only the introduction of novel theories or methods but also the disruption of existing paradigms. In this vein, our work challenges the prevailing neuroscience paradigm of using single genes to estimate neuronal activity. Our innovative approach, NEUROeSTIMator (freely available on GitLab), is unique in utilizing the entire transcriptome, which offers a more comprehensive and accurate measure of neuronal activity. To the best of our knowledge, our tool is the pioneer in this arena for RNA-seq based studies across tissue and individual cells.

The new main statement of innovation, now found on **line 57-62** is:

“Using a comprehensive multi-species gene expression dataset, our deep learning model employs a single unit bottleneck to derive interpretable neuronal activity scores and an auxiliary input to the decoder to counteract dataset-specific biases. This novel approach, which distills the whole transcriptome into a single integrative activity score, challenges the single-gene paradigm (e.g., Fos) for measuring activity at the single cell and bulk tissue level.”

We provide additional detail in the point-by-point responses below:

REVIEWER: Claim of Uniqueness: The authors assert that no other method has attempted this type of modelling. Although this might be true, providing more specific references or evidence could substantiate this claim.

RESPONSE: While numerous transgenic single-gene reporter systems exist to identify neuronal activity(see Hudson 2018⁷, Kawashima 2014⁸, Guenther 2013⁹, and Liu 2014¹⁰), they necessitate genetic alterations and are limited to model organisms. We underline that there is no existing tool designed to gauge neuronal activity using transcriptome-wide gene expression data from single cells.

REVIEWER: Data Collection: The authors emphasize the importance of their dataset's size and diversity. While data collection is critical to any study, it's not typically considered a methodological innovation.

RESPONSE: We value the reviewer's insight into the role of data collection in our study. While our work is anchored in machine learning, the foundation of our model's success rests significantly on the curated dataset. We outline the following aspects that support the curated dataset's contribution to the innovation of our work:

1. Generalization across species: Our curated dataset, comprising single cell gene expression data from multiple species' brains, is not merely about volume but breadth and depth. This

diversity ensures our model learns a more holistic representation, enabling broader applicability across varied contexts.

2. Mitigation of biases: While every dataset carries inherent biases, our careful curation across diverse species aims to mitigate these. Ensuring data reliability is crucial for preventing biases from skewing the model's predictions.

3. Dataset as driver of innovation: While there's undeniable value in refining model architectures, our belief is that a novel, well-curated dataset can also be a catalyst for groundbreaking results. It provides fresh insights that might be obscured in one-off datasets.

4. Dataset longevity: As the field advances, model architectures will evolve. However, a comprehensive dataset, once curated, can continue to serve as a bedrock for numerous future models, making its contribution enduring.

In essence, while the NEUROeSTIMator model architecture is central to our work, the underlying dataset is the bedrock upon which it stands. We believe this dual emphasis on data curation and model development collectively elevates the novelty and utility of our study. We hope this clarifies the pivotal role our data curation efforts play in the overall innovation of our research.

REVIEWER: Deep Learning Application: The authors mention using deep learning but could explain more thoroughly how they have applied it innovatively.

RESPONSE: Deep learning, with its inherent flexibility, has been a transformative force in multiple domains. In our study, we've leveraged this capacity in two notable ways: the single unit bottleneck and an auxiliary input to the decoder, both of which are tailored to our specific challenges in measuring neuronal activity.

1. Single unit bottleneck for interpretable scoring:

- Purpose: In many deep learning applications, the latent space can be multi-dimensional and challenging to interpret. With our single unit bottleneck, we have intentionally constrained the complexity of this latent space.

- Innovation: By doing so, we compel the model to distill vast amounts of data into a singular, interpretable score for neuronal activity. This score, being derived from a highly complex input, embodies a condensed yet comprehensive representation of neuronal activity. It's akin to extracting the 'essence' of the input data, providing an elegant and efficient method to gauge neuronal activity from the whole transcriptome.

- Advantage: This makes the results not only computationally efficient but also readily interpretable by researchers, enhancing both the usability and interpretability of our model. We also investigated, at the behest of reviewer 2, the possibility of a 2- or 3-unit bottleneck, neither of which showed an improvement on this task over the single unit bottleneck (see response below for further details).

2. Auxiliary input to the decoder for bias mitigation:

- Purpose: In diverse datasets like ours, spanning multiple species and contexts, there's a risk of the latent space inadvertently capturing dataset-specific biases. These biases could skew the neuronal activity score, reducing the generalizability and accuracy of the results.

- Innovation: The auxiliary input to the decoder serves as a protective mechanism. Instead of letting the single unit bottleneck learn and represent these biases, the decoder captures and processes dataset-specific, cell type-specific, and QC-related information separately.

- Advantage: This bifurcation ensures that the primary latent space remains unpolluted by external biases, focusing solely on neuronal activity. It's a strategic modification of the traditional autoencoder architecture, ensuring that our model's outputs are not only accurate but also consistently unbiased across varied datasets.

In summary, our approach to deep learning is tailored, considering the unique challenges posed by neuronal activity estimation from gene expression data. Through innovations like the single unit bottleneck (now further justified in the response to reviewer 2) and the auxiliary input to the decoder, we've crafted a model that marries computational robustness with practical interpretability and bias mitigation. These innovations are rooted in our commitment to advancing the understanding of neuronal activity in the most rigorous and nuanced manner possible.

REVIEWER: Auxiliary Decoder Branch: A clearer explanation, possibly accompanied by a figure or diagram, could help clarify this aspect of their methodological innovation.

RESPONSE: We agree with the reviewer that the auxiliary decoder could be described in more detail. In our revision, we have expanded the details of this facet of the model architecture and produced a new graphical representation of the model architecture (see **lines 598-603 and Supplementary Figure 10a**). To summarize, the auxiliary input to the decoder is a strategy to alleviate the latent space from pressure to learn dataset-specific, cell type specific, or QC-related information for the reconstruction of target gene expression. These three groups of variables represent some of the most profound sources of variation in single cell RNA-sequencing datasets. In early iterations of the model, we found that the latent space was susceptible to learning shortcuts to reducing loss by recognizing a certain cell type marker as associated with target gene expression if the corresponding cell type had higher target expression on average, even at baseline. We viewed this behavior as undesirable and introducing bias, as we would expect different datasets or cell types to exhibit comparable distributions of the latent space variable. The auxiliary input branch alleviates pressure to learn these biases by directly feeding dataset, cell type, and QC information into the architecture of the model after the latent space as part of the decoder. Feeding these inputs in downstream of the latent space is crucial, because nothing beyond the latent space is used in NEUROeSTIMator, and therefore the end user does not need to provide any additional metadata about their samples when they use our tool. To reduce the risk of overfitting, this auxiliary input branch encodes approximately 50 variables into a dense layer of six units before joining the latent

bottleneck layer to form the decoder for target gene reconstruction. This strategy is not novel in and of itself, though it is one of the ways we have modified the traditional autoencoder architecture to promote the learning of a latent space representation that robustly differentiates stimulated from non-stimulated neurons.

REVIEWER: In conclusion, while I appreciate the work that has gone into this manuscript, I believe that further revisions addressing these concerns (particularly the 1st major comment) are needed to sufficiently elevate its scientific contribution. I look forward to seeing how this work evolves and advances our understanding of neural activity.

RESPONSE: We have comprehensively addressed the reviewer's comments. First, we have taken the recommendation to reconstruct the whole transcriptome during the training of our model. The results, shown in an updated **Supplementary Figure 7-8** and described in **lines 386-389, 613-615 & 738-740**, show a *reduction* in performance (average delta AUC -0.03) compared to our target-gene-only reconstruction. Consequently, we are unable to find evidence that reconstructing the transcriptome is an effective approach for the task at hand. Second, we have further elucidated the specific innovative aspects to our work, more concisely in the manuscript and more comprehensively in the response to the reviewer's comments. These comments have strengthened the manuscript, and we hope that the reviewer finds these good-faith efforts satisfactory.

Reviewer #2 (Remarks to the Author):

REVIEWER: I appreciate the authors' detailed responses, which are not mere verbiage, but accompanied by time-consuming and thoughtful analysis and fairly deep reconfiguration of their model. In particular admire your CCA, additional test of the triple intersection for selecting genes, and comments on the impact of glial proportions were slick. You've clearly got a diversely-talented yet integrated team working on this.

RESPONSE: We sincerely thank the reviewer for their continued interest in our work, and for their insightful comments in this review.

REVIEWER: There is one essential remaining point that's been brought up repeatedly, that has not been reasonably addressed. That is the issue of a single dimension bottleneck. The rationale for this is summarized by the authors below, with my comment in brackets. Generally I thought this justification lacked the data support and high rigor of all other responses.

“A single score is easier to understand [actually reviewers are having a harder time understanding it because it is so limited as to be difficult to square with ephys], visualize [millions of informative 2d plots published each year], compare across samples [we routinely compare many dimensions across samples and, anticipating your response, while data reduction is helpful, I don't think anyone will balk at a couple of variables], and to integrate into downstream analyses such as classification or regression (e.g., as a covariate) [actually it looks straightforward]. While a bottleneck with additional capacity would undoubtedly capture more complex patterns [we don't know if they will be complex or not], it would also become more challenging to interpret the relationships between multiple dimensions [you don't know what without testing, an frankly anything is more challenging than no comparison (1D), but could also supply information that is useful], thus adversely impacting the tool's ease of use in the intended applications [we live in a world of 100k-variable datasets popping out every day, so going from a 1d to 2d model cannot realistically be an objection].”

RESPONSE: The reviewer has made some excellent points challenging our rationale for only considering the single-unit bottleneck. In response to these comments, we have built new models containing 2 and 3 units in the bottleneck for comparison to the single-unit bottleneck. Details of these new analyses can be found below, as well as on lines 389-391 & 741-744 and in Supplementary Figure 7-8.

REVIEWER: I would suggest that exploring 2 and 3-node inner layers of your encoder is actually not contrary to your current position - if you try that and find it is not useful for various reasons, then you have actual support for the current model. If that's the case, in the future paper, you just have a sentence like: We explored a large bottleneck but found

problems xyz. However, if you do make it work, you're opening the door to a more general methodology for inter-relating multi-scale systems vs more of a denoising mechanism (I'm simplifying, but I think you get it). This question is entwined with the findings that it takes 15 PC's to hit 90% of ephys variance and PC1 only providing 15% of the variance of your gene set. That latter is portrayed as high, but actually it's quite low for a set of genes thought to be functionally related. Both of these point to the potential for adding a few nodes to the bottleneck. I hope you can see from my prior questioning, I think this work has great potential, and I'm not just asking for stuff as an exercise, but think it's fundamental to realizing the direction on which you've already embarked.

RESPONSE: As mentioned above, we have built two new models containing 2 and 3 units in the bottleneck (see lines 389-391 & 741-744 and Supplementary Figures 7-8). Ultimately, we concluded that the additional capacity in the latent space was not particularly useful for multiple reasons. First, we found that in both the 2-unit and 3-unit models, activation outputs of the bottleneck units were highly correlated (test set correlations between all unit-unit pairs were greater than 0.9). These high correlations suggest that the latent space is not benefitting from the increased capacity of the network to reconstruct gene expression of the target genes. To summarize the outputs to a single score for the benchmarking analysis, we used the training set to fit a generalized linear model to predict stimulation status from latent variables and applied it to the test set. These two models showed the highest performance of all competing methods in our benchmark analysis, though our original model marginally outperformed them with a mean AUC increase of 0.007 for the 2-unit model and 0.015 for the 3-unit model (0.725 and 0.717 vs. 0.732 for the single unit model). We also tried comparing performance to each individual unit from these two models, but none of those units outperformed our original model.

In summary, the reviewer's invitation to investigate alternative 2- and 3- unit bottlenecks has strengthened the manuscript and in particular the justification for using a single unit bottleneck. Where before we relied on an appeal to intuition for that design choice, we now can show empirically that higher-dimensional bottlenecks do not offer superior performance. We appreciate the reviewer's encouragement to explore this possibility.

Reviewer #3 (Remarks to the Author):

REVIEWER: The authors have made substantial improvements to the manuscript, in both the revisions to the text and additional analyses. Although the variance explained still remains small using a glmnet model (~5.4%) and the AUC improvements are somewhat modest, the results and models presented do represent an important step forward in understanding the relationship between the cellular transcriptome and its neuronal activity. I expect this manuscript to be well cited, and look forward to its reception and impact on the field upon publication.

RESPONSE: We thank the reviewer for their comments here and in the previous review. We believe the changes we made in response to the previous review have substantially improved the quality of our manuscript.

REVIEWERS' COMMENTS

Reviewer #1 (Remarks to the Author):

Having reviewed the revisions made by the authors in response to my previous comments, I am pleased to note that all my concerns have been comprehensively addressed. I appreciate the efforts of the authors in making the necessary changes, and I believe that their revisions have enhanced the quality of the manuscript. Consequently, I recommend the acceptance of this manuscript for publication

Reviewer #2 (Remarks to the Author):

Dear Authors,

After this latest round of review and the extensive additional testing you have documented, I'm reasonably convinced of the robustness of the architecture and resulting biological conclusions. In particular, I appreciate the extensive comparisons to other methods, and exploration of alternative architectures, as I think this contributes to the broader need for understanding principles of DNN design, and of course, it's rather involved to do so.

Regards,
Chris